



# Rainfall pattern greatly affects water use by Mongolian Scots pine on a sandy soil, in a semi-arid climate

HongZhong DANG[1,a], LiZhen ZHANG[2], WenBin YANG[1], JinChao Feng[1], Hui HAN[3], Wei LI[1]

[1] Institute of Desertification Studies, Chinese Academy of Forestry, Beijing, 100091, China

[2] Institute of Resources and Environment, China Agricultural University, Beijing, 100094, China

[3] Institute of Sand Fixation and Afforestation of Liaoning Province, Fuxin, 123000, China

[a] *Correspondence to*: HongZhong DANG (hzdang@caf.ac.cn)

**Abstract.** We report new information on tree water use by Mongolian Scots pine (*Pinus sylvestris* var. *mongolica*) growing
on a sandy soil, in a region characterised by an erratic rainfall pattern. Measurements were made over three successive years
of contrasting annual rainfall - a wet year (2013), a dry year (2014), and a second dry year (2015). The result was the development of worsening levels of drought year by year. Over the three years, sap flux density ($J_s$) was measured at individual tree
level in up to 25 trees. The sap flux density values were up-scaled to estimate tree water use at plot level ($T_s$). Our measurements
follow forest plot response to increasing levels of drought which developed over a three-year period as soil moisture conditions
gradually worsened from wet, to moderate-drought, to severe-drought, to extreme-drought, in response to the dynamics of a
variable rainfall pattern. Values of $T_s$ did not exceed 3.03 mm day$^{-1}$ (2013), 1.75 mm day$^{-1}$ (2014) and 1.59 mm day$^{-1}$ (2015)
during the three growing seasons. Total annual stand transpiration over the same three years declined progressively from 290
mm (2013), to 182 mm (2014) and to 175 mm (2015). Satisfactory power-function relationships ($R^2 = 0.64$) between daily $T_s$
and the product of ET$_0$ and the relative extractable soil water (REW) were found. This study helps elucidate the interplay
between the effects of the atmosphere and soil moisture on tree water use. Tree water use responded to drought, with daily $T_s$
values decreasing by 5–46% in response to moderate drought, by 48–62% in response to severe drought and by 65% in response to extreme drought. Upon release of moderate drought by heavy rainfall in 2013, daily $T_s$ recovered completely. However, under the severe and extreme droughts in the subsequent dry years, recovery of $T_s$ following heavy rainfall was incomplete (57–58%). Our results highlight the negative effects of water stress on the growth of mature forest trees, in a sandy soil,
in a climate characterised by large intra- and inter-annual variances in rainfall. When the erratic rainfall and sandy soil were
also coupled with a declining groundwater table, the result was tree water use fluctuated widely over quite short time scales
(months or weeks). Overall, our findings account for the observed premature degradation of these MP plantations in terms of
an eco-hydrological perspective.

**Keywords:** sap flux; Mongolian Scots pine (*Pinus sylvestris* var. *mongolica* Litv); soil water availability; water stress; sandy
soils; semi-arid climate.



## 1  Introduction

Reforestation has been used widely in semi-arid areas to control soil erosion, to capture carbon and to serve as wind breaks (D'Odorico and Porporato, 2006). Nevertheless, trees growing in severely water-limited ecosystems are often exposed to significant challenge due to insufficient soil water (Wesche et al., 2011; Su et al., 2014). Many factors influence the amount of soil water and its availability to vegetation. For instance, the amounts and timings of rainfall, soil water capacity, root water-uptake capacity and the availability of alternative water sources such as groundwater (Meinzer et al., 2006). Based on current climate-change forecasts, the increasing in the frequency and severity of droughts and thereout decreased soil water availability are likely to become more common in future (Leo et al., 2013). This will tend to increase tree mortality rates through excessive competition for water and thus will influence the structure and functioning of forest ecosystems (Barbeta et al., 2015). Therefore, quantification of water use by individual trees and forest stands, and increased understanding of how environmental factors affect their water usage, are very necessary if we are properly to assess the impacts of climate change on ecological and hydrological processes in these fragile ecosystems (Bovard et al., 2005). Better understanding of these issues will also allow us to make better forest establishment decisions and to improve our forest management actions.

Mongolian Scots pine (*Pinus sylvestris* var. *mongolica*, MP), a geographical variety of Scots pine (*P. sylvestris*), is naturally widely distributed in northern China and in parts of Russia and Mongolia. For example, it is found in the Daxinganling Mountains (50°10′–53°33′N, 121°11′–127°10′E) and in Honghuaerji on the Hulun Buir sandy plains of the northeast (47°35′–48°36′N, 118°58′–120°32′E)) (46°30′–53°59′N, 118°00′–130°08′E) (Zhu et al., 2008; Zheng et al., 2012). The MP is a popular species for reforestation in northern China due to its traits of good drought and cold resistance. Consequently, more than $6.7 \times 10^5$ ha of MP plantations have been established to control desertification, in the great project of the Three-North Shelter Forest Program (TNSFP) launched in China in 1978 (Zheng et al., 2012). Unfortunately, serious degradation has occurred in these plantations since the mid-1990s. This has included poor tree health and also tree death, particularly in those plantations on the sandy soils of southern Horqin (42°43′N, 122°22′E, our study area). This is causing considerable concern (Jiao, 2001; Zhu et al., 2008).

A key driver for the degradation of these water-limited ecosystems is obviously the region's low and erratic rainfall and its effect on soil water availability (Mereu et al., 2009). In our semi-arid research site, with its sandy soils, any combination of the three main soil-water related factors could be the problem: the high inter-annual variation in precipitation, the high intra-seasonal variability in precipitation and the declining groundwater table. However, it remains unclear how, and to what extent, these three factors are actually responsible for the degradation recorded in our site. This is because the forest's sensitivity to drought will be highly species-specific, highly climate-specific and highly site-specific. In particular, few studies have been conducted with MP trees, growing in these sandy soils, and which involve large numbers of sap flux measurements, made in multiple trees and across several growing seasons.

In general, plants native to arid and semi-arid environments have developed a wide range of water-use strategies to cope with drought. One such strategy, employed by 'water saver' plants, is to avoid drought by minimising their water use by limiting leaf area development, by defoliation or by stomatal closure etc (Levitt, 1980; Gartner et al., 2009; Chirino et al., 2011). MP is a shallow-rooted species with over 85 % of roots located in the upper 0.4 m of the profile and sharply decreased root density with depth down to 1.0 m in our site (Su et al., 2006). We thus hypothesise that the high dependence of MP trees on the moisture contained in the upper soil layers is the main reason for degradation of these forest stands under conditions of prolonged drought.

In this study, we measured the sap flux density ($J_s$) on up to 25 trees in a stand of MP, semi-continuously over three consecutive years - a wet year followed by two successive dry years. We did this using thermal dissipation probes (Granier, 1985). We also made concurrent measurements of the key environmental variables including rainfall ($R$), volumetric soil water content ($\theta$) and the level of the groundwater table ($g_w$). The aims of this study were: (1) To record changes in the daily water use of an MP stand, based on sap-flux measurements made across three contrasting rainfall years; (2) To determine the relationships between $J_s$ and the main meteorological variables, as well as in relation to changes in soil water availability which occurred as the intensity of a drought deepened over the extended (three-year) period; (3) To quantify the severity of the drought stress by exploring variances in the rate of daily water use, over a number of cycles of soil wetting and drying.




## 2 Materials and methods

### 2.1 Site description

The trial was carried out at the Zhanggutai National Desertification Control Trial Station located on the southern edge of the Horqin region of sandy soils in Liaoning province, China (122° 22′ E, 42° 43′ N, at 226.5 m a.s.l.). The selected plot is in a 40-ha plantation of 35-year old MP, planted at a density of about 625 trees per ha. Management interventions and other human disturbances were limited by the installation of a secure fence around the plot. The site has a semi-arid, continental climate with a mean annual temperature of 7.9°C, a frost-free period of 150–160 days per year, a mean annual evaporation of 1553.2

mm and a long-term annual mean precipitation of 474.7 mm ($P_{ave}$) over the last 30 years (1983–2012) with coefficient of variance of 0.27 (Zhu et al., 2005). Over the longer period, there have been a number of consecutive dry years. For instance, annualprecipitation values between 1996 and 2004 were well below $P_{ave}$ (**Fig. 1a**). Usually, about 60 to 70 % of annual rainfall occurs in the three months between June and August. The value of annual temperature over the last 30 years has increased slightly, at a rate of about 0.03°C yr$^{-1}$ (**Fig. 1b**), while annual precipitation has decreased slightly, a rate of about 2.0 mm yr$^{-1}$.

The soil is of an aeolian sandy type with the bulk density of 1.61 g cm$^{-3}$. The soil texture is of 83 % sand (> 0.05 mm), 9 % silt (0.05–0.002 mm) and 8 % clay (< 0.002 mm) (Jiao, 1989). The sub-canopy species in the plantation area include *Acer pictum* subsp. *mono* Maxim, *Crataegus pinnatifida var. major N. E. Brown.*, *Lespedeza bicolor* Turcz., *Artemisia halodendron* Turcz et Bess., *Cleistogenes chinensis* Maxim. Keng et.al.

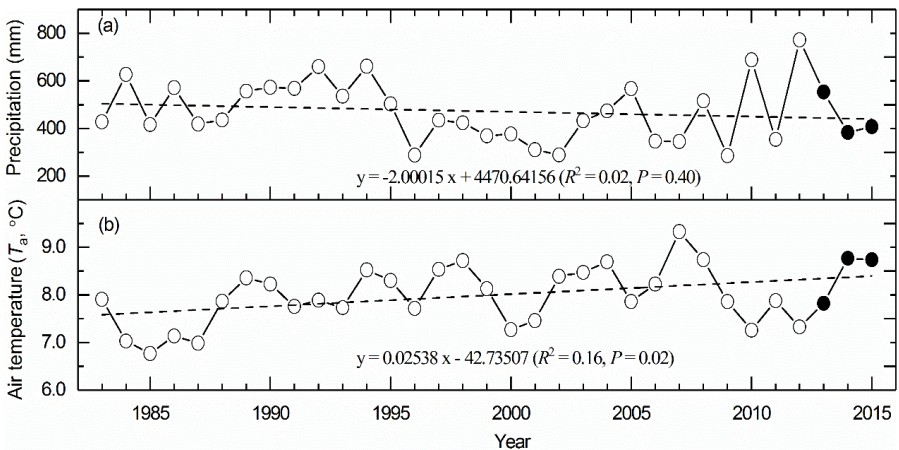

**Figure 1** Inter-annual variations in (a): mean annual air temperature, (b): annual precipitation at Zhanggutai Station. Open circles are the values recorded over the previous 30-year period (1983-2012). Solid circles are the values recorded during the three-year period of this investigation (2013-15). The dashed lines indicate the linear regressions over the period for the two variables.

### 2.2 Trial plots and sample trees

To break the prevailing northerly winds, the MP are planted in a square-grid pattern with an east-west, north-south orientation and tree spacing about 4×4 m. A sample plot of size 20×20 m (~ 0.04 ha) containing 25 trees, was established in this even-aged, monoculture plantation and was surrounded by a wire fence (**Fig. 2**). The numbers of trees instrumented for sap flux measurements were 25 in 2013, 22 in 2014 and 13 in 2015. The decreasing numbers of instrumented trees was due to progressive sensor malfunction and/or cable damage by squirrels and other rodents. The main characteristics of the sampled trees are

shown in **Table 1**. Diameters at breast height (DBH) were measured with a diameter tape and tree height with an altimeter. Trees fitted with sapflow sensors were also core sampled with a Pressler increment borer at breast height to determine the thicknesses of bark, sapwood and heartwood. Thickness measurements were made with a Vernier caliper with tissue boundaries identified based on color. The DBH, tree height and sapwood areas of the sampled trees in 2013, 2014 and 2015 were not





significantly different at significance of 0.05 (the $P$ values for three    years were 0.43, 0.06, and 0. 39, respectively).

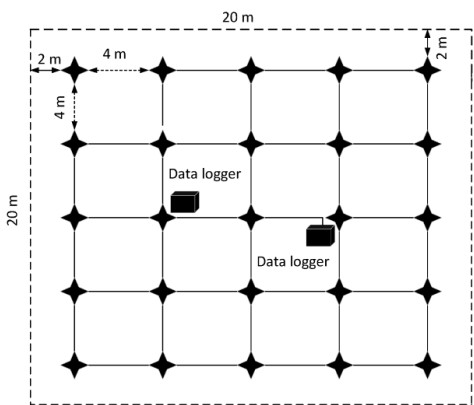

**Figure 2**  Sketch map of 25 sample trees (stars) planted in a 4×4 m spaced square grid of about 400 m$^2$ (dashed line is border fence). Tree ages were identical and tree sizes were similar. The number of instrumented trees decreased in 2014, and again 2015. Details of samples see **Table 1.**

**Table 1** The main attributes of sampled trees with means and standard deviations (S.D.), including diameter at breast height (DBH; cm), tree height ($H$; m), height of first live branch ($H_b$; m), 1$^{st}$ quartile of DBH ($Q_1$), 3$^{rd}$ quartile of DBH ($Q_3$), sapwood area ($A_s$, cm$^2$), and the number of sampled tress ($n$).

| Year | DBH / cm | | | | $H$ / m | $H_b$ / m | $A_s$ / cm$^2$ |
| --- | --- | --- | --- | --- | --- | --- | --- |
| | mean ± S.D. | Max-Min | $Q_1$ | $Q_3$ | | | |
| 2013 ($n = 25$) | 18.04 ± 2.65 | 24.7–12.9 | 16.1 | 18.9 | 10.28 ± 0.66 | 4.51 ± 1.11 | 202.61 ± 58.61 |
| 2014 ($n = 22$) | 17.14 ± 2.13 | 21.1–12.9 | 15.6 | 18.7 | 9.26 ± 0.81 | 3.67 ± 0.46 | 182.32 ± 41.98 |
| 2015 ($n = 13$) | 17.70 ± 2.19 | 19.86–12.9 | 17.2 | 18.8 | 9.05 ± 0.84 | 3.70 ± 0.34 | 188.61 ± 52.26 |

### 2.3  Micrometeorological variables

Micrometeorological variables including solar radiation ($R_s$), net radiation ($R_n$), air temperature ($T_a$), relative humidity (RH), wind speed ($W_s$) and rainfall ($R$) were measured using an automatic weather station (AR5, Avalon Scientific, Inc. USA) located about 50 m away from the plot. All sensors were installed 2.0 m above the ground except the rain gauge, which was 0.5 m above the ground. Variables were measured at 1 min intervals and their average values recorded every 10 min. Reference evapotranspiration (ET$_0$) was calculated using the FAO Penman-Monteith equation based on the variables $R_n$, $T_a$, RH and $W_s$ sourced from the weather station (Allen et al., 1998) (Eq. (1)). Hourly ET$_0$ was summed to daily ET$_0$.

$$ ET_0 = \frac{0.408(R_n - G) + \gamma \frac{900}{T_a + 273} u_2 (e_s - e_a)}{\Delta + \gamma(1 + 0.34 u_2)} \qquad (1) $$

where ET$_0$ = reference evapotranspiration (mm h$^{-1}$),

$\Delta$ = slope of saturated water vapour pressure against air temperature $T_a$ (kPa °C$^{-1}$),

$R_n$ = net radiation (MJ m$^{-2}$),

$G$ = soil heat flux (MJ m$^{-2}$),

$\gamma$ = the psychrometric constant (kPa °C$^{-1}$),

$e_s$ = saturated vapour pressure (kPa),

$e_a$ = actual vapour pressure (kPa), and

$u_2$ = mean wind speed at 2 m height (m s$^{-1}$).

The value of $D$ (kPa) was calculated using the following formula (Campbell and Norman, 1998):

$$ D = 0.611 exp^{\left(\frac{17.502 T_a}{T_a + 240.97}\right)} (1 - RH) \qquad (2) $$



### 2.4 Soil moisture and groundwater table

Volumetric soil moisture contents ($\theta$, %) were measured at depths of 0.1, 0.2, 0.4, 0.6, 0.9, 1.2, 1.6 and 2.0 m (ECH$_2$O EC-5 sensors, Decagon Devices Inc., USA). Sampling was at 10 min intervals with hourly means recorded by a SQ2020 data logger (Grant Instruments Ltd, UK). The sensor-specific measurement data were calibrated using a site-specific equation based on the oven-drying method (Eq. (3)):

$$\theta = 0.9677\theta_s + 0.2635 \quad (R^2 = 0.96, n = 194, \mathrm{RMSE} = 0.41) \tag{3}$$

where $\theta_s$ are the raw values; $\theta$ is the calibrated soil water content indicated at each depth - for example, $\theta_{1.0\,m}$ is the value of $\theta$ at 1.0 m. The weight-averaged value of $\theta$ was calculated over a range of depths - for example, $\overline{\theta}_{0-1.0\,m}$ is the weight-averaged value of the measurements from 0 to 1.0 m depths. The mean field capacity ($\theta_f$) of this soil is 18 % based on field measurements. The minimum value of $\overline{\theta}_{0-1.0\,m}$ ($\theta_{min}$) over the three years of the trial was 2.3 %. Relative extractable water (REW) in the upper 1.0 m of soil was calculated using Eq. (4) (Granier, 1987).

$$\mathrm{REW} = \frac{\overline{\theta}_{0-1.0\,m} - \theta_{min}}{\theta_f - \theta_{min}} \tag{4}$$

The more specific classification to quantify the degree of drought at our site is defined in **Table 2**.

Groundwater table ($g_w$) was monitored *in situ* manually once per month using a measuring tape with a cone.

**Table 2** Classification standard of the degree of drought (SWD) based on the relative extractable water (REW) and the mean soil moisture in the upper 1.0 m soil ($\overline{\theta}_{0-1.0\,m}$) (Jiao, 2001; Zhu et al., 2005; Song et al., 2016) in this site and our results in Fig. 6

| Labels | REW levels | Volumetric soil water content (%) | Degree of drought |
|--------|-----------|-----------------------------------|-------------------|
| SWD$_0$ | REW > 0.43 | $\overline{\theta}_{0-1.0\,m}$ > 9.0 % | No drought |
| SWD$_m$ | 0.24≤ REW ≤ 0.43 | 6.0 % ≤ $\overline{\theta}_{0-1.0\,m}$ ≤ 9.0 % | Moderate drought |
| SWD$_s$ | 0.08 ≤ REW < 0.24 | 3.5 % < $\overline{\theta}_{0-1.0\,m}$ < 6.0 % | Severely drought |
| SWD$_e$ | REW < 0.08 | $\overline{\theta}_{0-1.0\,m}$ ≤ 3.5 % | Extreme drought |

### 2.5 Sap flux measurements

Sap flux density ($J_s$, cm min$^{-1}$) in the outermost (0–3 cm depth) sapwood was measured continuously using the thermal dissipation method (Granier, 1985). Each probe was installed on the north side of the stem at breast height (1.3 m) with pairs of probes 0.15 m apart vertically. The upper probe included a heater and the lower probe was unheated and so remained at trunk temperature for reference. Each sensor was carefully removed at the end of each growing season. The temperature difference between the upper (heated) probe and the lower (reference) probe was measured at 1-min intervals, with mean values recorded at 10-min intervals using SQ2020 data loggers and hourly average values being calculated. The sensors were shielded with thick aluminum-faced foam to minimise warming by radiation and exposure to rain and to physical damage. The Granier empirical equation for $J_s$ was adopted as Eq. (5):

$$J_s = 119 \times 10^{-4} \left(\frac{\Delta T_0 - \Delta T}{\Delta T}\right)^{1.231} \tag{5}$$

where $\Delta T$ is the actual temperature difference observed between heated and reference probes, and $\Delta T_0$ is the maximum $\Delta T$ value when sap flow is close to zero (generally just predawn) determined over about 10 consecutive days (Lu et al., 2004; Dang et al., 2014).

To upscale the sap flux density measurements $J_s$, to obtain an estimate of the water consumption of the stand per day, mean values of $J_s$ were calculated at 1 min intervals. These means were multiplied by sap wood area ($A_s$), divided by ground area ($A_g$) and summed over 60 min and then over 24 h to obtain daily sap flux of individuals per ground area ($J_t$, mm day$^{-1}$). The daily mean $J_t$ from all measured trees in the plot was used to as the surrogate of daily transpiration of the stand ($T_s$, mm day$^{-1}$). Because trees were of the same age and regularly-spaced, they were assumed to occupy equal areas of ground per sapwood area. Hence, the ground area fraction of each tree ($A_{g,i}$, cm$^2$) was approximated as the product of individual sapwood area and the ratio of total stand sapwood area ($A_{s\text{-}stand}$, m$^2$) divided by total stand ground area ($A_{g\text{-}stand}$, m$^2$) (Eq. (6),(7), (8)):




$$T_s = \frac{1}{n}\sum_{i=1}^{n} J_{t,i} \qquad (6)$$

$$J_{t,i} = \sum_{j=1}^{24} J_{s,i} \frac{A_{s,i}}{A_{g,i}} \times 60 \qquad (7)$$

$$A_{g,i} = A_{s,i} \times \frac{A_{s-stand}}{A_{g-stand}} \qquad (8)$$

where $J_{s,i}$ is the 10-min mean sap flux density of a tree $i$, $A_{s,i}$ is sapwood area of tree $i$, $A_{g,i}$ is ground area of a tree $i$ weighted by sapwood area, $J_{t,i}$ is the daily sap flux of a tree $i$ (mm day$^{-1}$), $n$ is the numbers of trees measured each year. The radial variance in sap flux density through the sapwood was not measured.

**2.6  Sapwood area**

Sapwood width was determined at the end of the growing season each year by core sampling and tissue color difference (see above) at the position of probe installed. In our Mongolian Scots pine, the sapwood colour was yellow-white and that of the heartwood was tan. A few drops of methyl orange solution helped define the interface where the boundary was indistinct. Sapwood area ($A_s$, cm$^2$) was calculated as the product of sapwood thickness and perimeter (**Table 1**). Volumetric estimates of
sap flux (cm$^3$ h$^{-1}$) were the product of sap flux density and sapwood area.

**2.7  Statistical analyses**

The metabolic activity of MP in this region usually starts in late April and ends in late October (Jiao, 1989), thus we defined the period from 1 May through 31 October as the growing season. Effects of atmospheric factors (lumped into ET$_0$ in this study) on daily $T_s$ under conditions of different levels of soil water were investigated by linear regression. The coupled rela-
tionships of ET$_0$ and $\theta$ on daily $T_s$ was fitted by a non-linear regression function. The ratio of stand transpiration to reference evapotranspiration ($T_s$ / ET$_0$) was calculated to describe the driver efficiency of transpiration from the atmosphere. The degree to which $g_w$, REW and $D$ controlled $T_s$ / ET$_0$ was determined by stepwise linear regression. The effect of soil water level on the ratio of $T_s$ / ET$_0$ was tested by one-way analysis of variance (ANOVA) and a Tukey HSD *post hoc* multiple comparisons test using SPSS 20 (SPSS Inc., Chicago, IL, USA). Significant correlations between $T_s$ or $T_s$ / ET$_0$ and environmental
factors over different periods were determined by Pearson's correction coefficient tests at $P < 0.05$ or 0.01. The other statistical analyses and plots employed OriginPro 2016 version 9.3 (OriginLan Inc., Northampton, MA, USA).

**3  Results**

**3.1  Seasonal dynamics of stand transpiration and environmental factors**

The amounts of rainfall ($R$) during the investigation periods were 554.2 mm (in 2013), 383.6 mm (in 2014) and 408.0 mm (in
2015). These were equivalent to 117 %, 81 % and 85 % of $P_{ave}$, respectively. Only 6 to 7 % of days in a year experienced heavier rainfalls greater than 10 mm day$^{-1}$, while between 25 and 29 % of rainy days had lighter rainfalls of less than 10 mm day$^{-1}$. In contrast, more than 65 to 67 % of days in a year were rainless. The cumulative rainfall amounts occurring during the 6 to 7 % of days of heavier rainfall was equivalent to 91 % of total annual rainfall in 2013, 78 % in 2014 and 68 % in 2015. Hence, rainfall tended to be concentrated over quite short periods. The maximum daily rainfalls in each of the three years were
79.8 mm on 2 July 2013, 45.6 mm on 6 June 2014 and 56 mm on 19 August 2015 (**Fig. 3a**). The rainfall patterns in this region were not such as to maintain soil moisture at a constant level nor at a high available level.

Daily soil water content exhibited large variances from day to day, which reflect very rapid refilling from rainfall events. The heterogeneity of soil water content with soil depth was also significant ($P < 0.001$). In the wet year, 2013, with ample rainfall, soil water content was higher with mean $\overline{\theta}_{0-0.6\,m}$ of 10.4 ± 1.9 %, $\overline{\theta}_{0.6-1\,m}$ of 8.6 ± 1.7 %, $\overline{\theta}_{1-1.5\,m}$ of 6.0 ± 1.7 % and
$\overline{\theta}_{1.5-2\,m}$ of 4.1 ± 0.8 %. In contrast, in the dry year, 2014, with a rainfall reduction by 170 mm, soil water content decreased to 7.5 ± 2.6 %, 5.3 ± 2.2 %, 3.3 ± 0.7 % and 3.0 ± 0.3 % for the same four soil layers (**Fig. 3b**). In the second dry year, 2015, the drought continued, with 7.3 ± 3.2 %, 5.4 ± 2.2 %, 3.0 ± 0.4 % and 3.3 ± 0.4 % for the same four soil layers, till late in August (DOY 232) when a heavy rainfall event (56.2 mm) wetted the whole soil layer. Based on our classifications (**Table 2**), the days of severe and extreme drought (SWD$_s$ + SWD$_e$) accounted for 20 %, 36 %, 64 % and 80 % of the whole three-year period




for the four soil layers at 0–0.6, 0.6–1, 1–1.5 and 1.5–2 m, respectively. Thus, for MP it is the upper 1.0 m soil layer that

provides the main water source, having highest levels of soil moisture ($\overline{\theta}_{0-0.6\,m}$ and $\overline{\theta}_{0.6-1\,m}$). Intense rainfall that infiltrated

to, and thus helped recharge, the deeper layers of soil ($\overline{\theta}_{1.5-2\,m}$) were very rare.

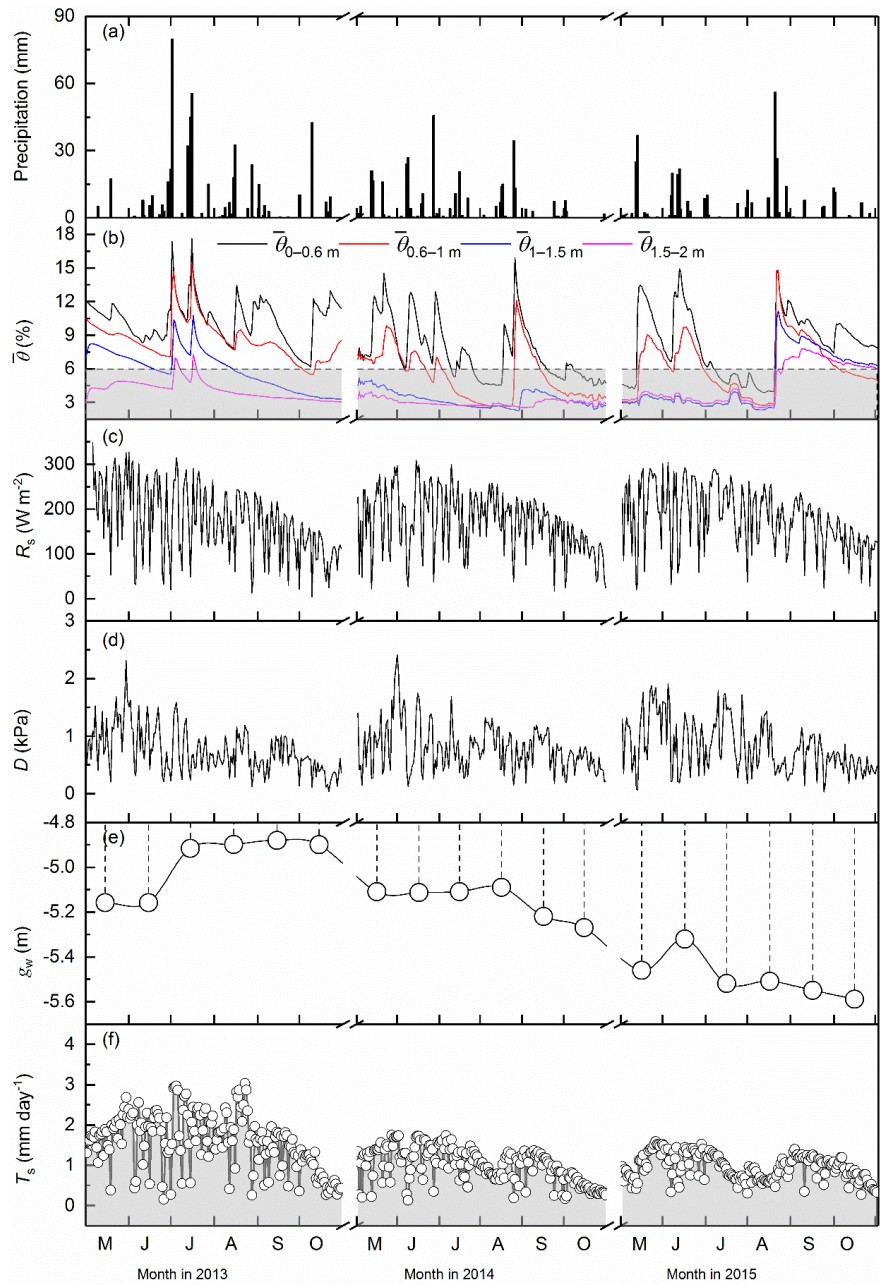

**Figure 3**   Seasonal time courses, over the three year period, of (a): rainfall ($R$), (b): volumetric soil water content

($\theta$), (c): solar radiation ($R_s$), (d): vapour pressure deficit ($D$), (e): groundwater table ($g_w$), (f): daily mean sap flux

of stands ($T_s$). The grey area in (b) was where $\theta$ was lower than 6.0%.

In general, the seasonal patterns of mean daily $R_s$ and $D$ over the three years are similar (**Fig. 3c, 3d**). There are no

significant differences between years for daily $R_s$ ($P = 0.4$) or daily mean $D$ ($P = 0.25$). The $D$ over the whole period never





exceeded 2.4 kPa (**Fig. 3d**). The groundwater table ($g_w$) at the start of the three year investigation was at about 5.2 m. This rose significantly to 4.9 m, following about 306.8 mm of heavy rainfall during June and July in 2013. However, $g_w$ then fell gradually in 2014 and 2015, down to 5.6 m at the end of the period. The annual mean values of $g_w$ over the three years were 5.0, 5.2 and 5.5 m, respectively (**Fig. 3e**).

The daily values of $T_s$ showed similar seasonal patterns to those of $R_s$ (**Fig. 3f**). Overall, $T_s$ increased gradually each year
from May, remained relatively high until September, then gradually decreasing to a low minimum in late October. The seasonal dynamics of $T_s$ was reflected in variations in a number of physiological traits of MP, which also reflected fluctuations in several meteorological factors. However, the values taken $T_s$ between the years were significantly different ($P < 0.001$). Some examples are: the maximum daily $T_s$ in 2013 was 3.03 mm day$^{-1}$ with an average of 1.58 mm day$^{-1}$; in 2014 this decreased to 1.75 mm day$^{-1}$ with an average of 0.99 mm day$^{-1}$; in 2015 this decreased further still to 1.59 mm day$^{-1}$ with an average of 0.94 mm
day$^{-1}$ (**Fig. 3f**). The decreasing values of $T_s$ between seasons was the result mainly of reducing rainfall but declining conditions of soil moisture were also involved.

### 3.2 Relationships between $T_s$ and $ET_0$

The daily values of $T_s$ were related to the daily $ET_0$ in the three years in the context of quite different soil moisture levels. Although good linear relationships between daily $T_s$ and daily $ET_0$ were found, the relationships became less strong - a reduced
slope coefficient and a reduced fit ($R^2$) - moving from conditions of no drought ($SWD_0$) (**Fig. 4a**), to moderate drought ($SWD_m$) (**Fig. 4b**), to severe and extreme drought ($SWD_s + SWD_e$) (**Fig. 4c**), where it can be seen that $ET_0$ explains about 59 % of the variance in daily $T_s$ for $SWD_0$, 45 % for $SWD_m$ and 29 % for $SWD_s + SWD_e$. These findings indicate that drought suppresses the effect of the atmospheric driver on sap flux. Meanwhile, the increase in sap flux with $ET_0$ is less steep with declining soil moisture due to an increasing resistance between soil and root for water absorption. Consequently, the difference in rainfall
level and thus in soil moisture level caused large and significant differences in daily stand transpiration ($P < 0.001$) with mean values (± S.D.) of 1.43 ± 0.66, 1.20 ± 0.52 and 0.83 ± 0.33 mm day$^{-1}$ for associated soil moisture levels: $SWD_0$, $SWD_m$, $SWD_s + SWD_e$, respectively.

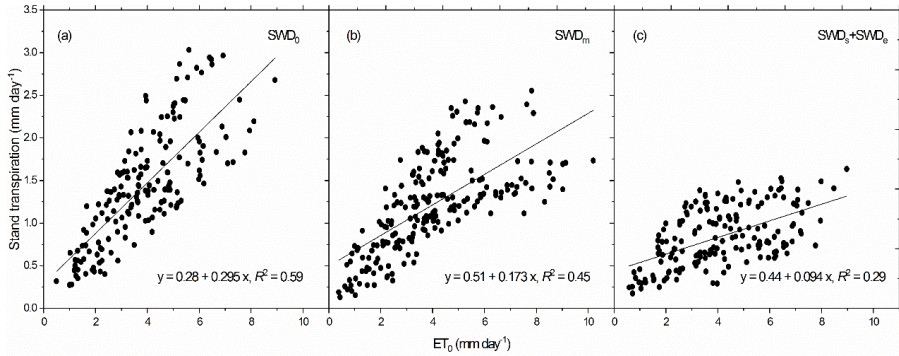

**Figure 4** Relationships between daily sap flux ($T_s$) and reference evapotranspiration ($ET_0$) under conditions of declining soil moisture. The degree of drought (SWD) based on the relative extractable water (REW) is (a): $SWD_0$ with REW > 0.43, (b): $SWD_m$ with REW: 0.24 - 0.43, (c): $SWD_s + SWD_e$ with REW <= 0.24, in the three years.

### 3.3 Combined effects of meteorological factors and soil moisture on sap flux

To help explain the residuals in the relationship between daily $T_s$ and $ET_0$, the relative soil moisture level was accounted for
by establishing a new variable EW where EW = $ET_0 \times$REW. The value of $T_s$ increased with EW following a power equation which explained about 64 % of the variance in $T_s$ (**Fig. 5**). Given the usual relationship between photosynthesis and transpiration, growth will be maximal under conditions of high water supply and high water consumption. In this fragile ecosystem, the common situation in summer is for frequent sunny days. Under these conditions soil water content is low and atmosphere



water demand is high and the result is mid-range values for $T_s$. The scatter in the plotted values about the fitted line (**Fig. 5**)

will likely be a result of the effects of a range of other factors, such as changing stomatal aperture.

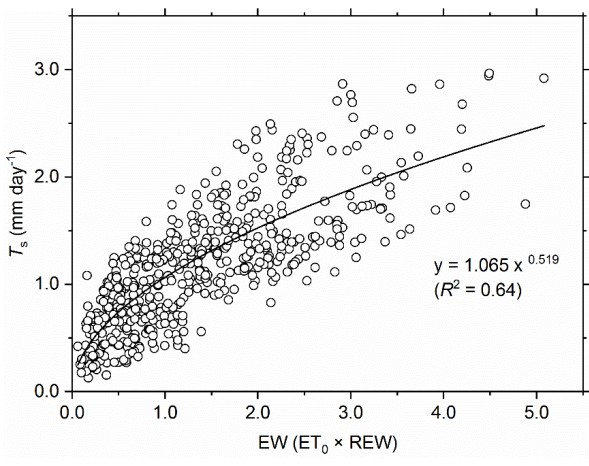

**Figure 5** Relationships of daily sap flux ($T_s$) varied with a new variable EW which is the product of relative extractable soil water in the top 1.0 m (REW) multiplied by reference evapotranspiration (ET$_0$) (EW = ET$_0$×REW).

**3.4 Progressive decline of sap flux with developing of drought and recovery following rain**

The 49-day periods in the same period each year (DOY 203 to 251) was chosen to illustrate changes in $T_s$ with the developing of drought as well as in dry-wet shift (**Fig. 6**). Overall, the mean $T_s$ during the same period in the three years were $1.80 \pm 0.63$,

$0.99 \pm 0.29$ and $0.82 \pm 0.27$ mm day$^{-1}$, respectively. In the wet year (2013), the mean $T_s$ in SWD$_0$ stage was about 1.82 mm day$^{-1}$ before the beginning of stress (**Fig. 6a**). Although the value of $T_s$ decreased by 5 % in the stage of SWD$_m$ (**Fig. 6a**), it recovered completely following heavy rain (**Fig. 6a**). This indicates a healthy response during dry-wet cycling. In the first dry

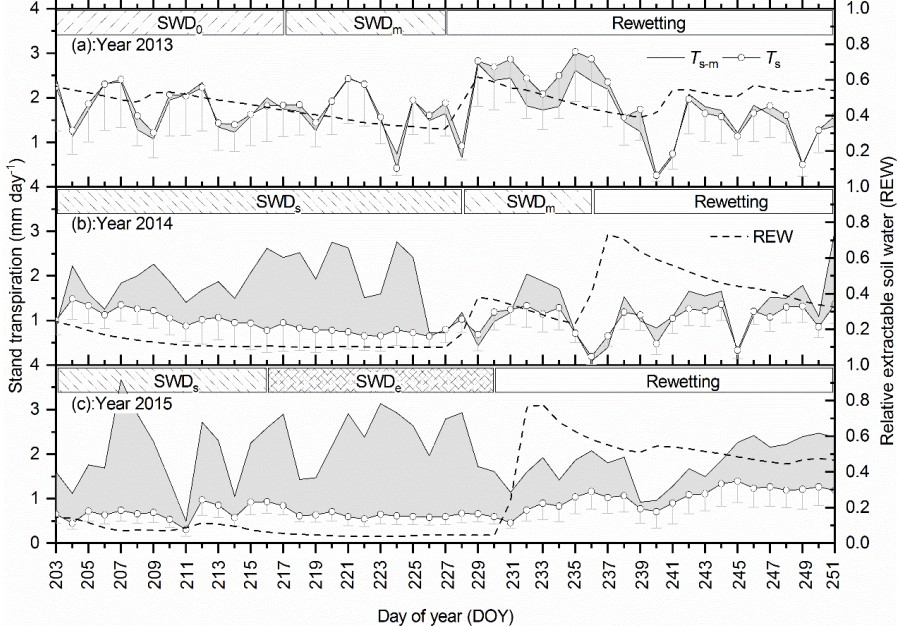

**Figure 6** The 49-day period (from DOY 203 to 251) of each year has been selected to illustrate changes in measured sap flux ($T_s$) during prolonged drought and also during dry-wet changes with relative extractable water in the top 1.0 m depth (REW) and modelled sap flux ($T_{s\text{-}m}$). The value $T_{s\text{-}m}$ was calculated based on the linear regression function of $T_s$ - ET$_0$ in SWD$_0$ stage (**Fig. 4a**). The grey area between $T_{s\text{-}m}$ and $T_s$ indicates their differences. The degree of drought, SWD$_0$, SWD$_m$, SWD$_s$, and SWD$_e$ are defined in **Table 2**. Soil moisture increased (rewetting) following heavy rain. For clarity, only half the S.D. of daily $T_s$ is shown.




year (2014), mean $T_s$ decreased by as much as 46 % in the stage of $SWD_m$ and further by 48 % in the stage of $SWD_s$. However, $T_s$ incompletely recovered following heavy rain to only 57.7 % of that in the stage of $SWD_0$ (in 2013) (**Fig. 6b**). In the second

dry year (2015), a similar trend but more great decreasing occurred with $T_s$ decreased by 62 % in the stage of $SWD_s$ and further by 65 % in $SWD_e$ stage. The daily $T_s$ in $SWD_e$ stage was only about 0.63 mm day$^{-1}$. This indicates there remained too little residual soil moisture (REW < 0.08) to supply for MP's normal behavior. Although, following heavy rain in late August 2015, $T_s$ recovered to nearly the same level as in 2014 (1.04 mm day$^{-1}$), it was still far lower than in the non-drought period (57.1 % of that in $SWD_0$ stage). Obviously, in both dry years, after long periods of severe drought, $T_s$ did not fully recover, even

following good rainfall and a much improved level of soil water.

## 4   Discussion

### 4.1   Harsh environmental conditions around the plantation

In our site, the long term trend for increasing air temperatures and decreasing annual precipitation (**Fig. 1**) which fits with the climate change prediction for the northern hemisphere (IPCC, 2007), is unfavorable to the growth of plantation. Meanwhile,

both the declining level of groundwater and the coarse sandy soil (>83 % sand particles in our site) prevented capillary ascension efficiently (less than 0.5 m)(see review by (Vincke and Thiry, 2008). Moreover, sandy soils also have low water holding capacity and high hydraulic conductivity so water percolates through quickly following rain. As a result, under well-wetted soil conditions ($SWD_0$), $\overline{\theta}_{0-1.0\,m}$ was depleted at the high rate of 1.9 vol % per day during the first two days and at the rate of 0.35 vol % per day during the subsequent nine days (**Fig. 7**). The depleting rate of $\overline{\theta}_{0-1.0\,m}$ under the drought conditions of

$SWD_m$ was 0.44–0.47 vol % per day, and decreased to only 0.09 vol % per day under the conditions of $SWD_e$. The rapidly depleted soil moisture is responsible for the frequent drought to which the time with degrees of moderate drought, severe drought and extreme drought accounted for about 38 %, 26 % and 3 % of the whole trial period, respectively. Consequently, the sap flux declines very quickly in this partially desiccated root system (Mereu et al., 2009).

### 4.2 Transpiration contribution from the upper soil layers

The relative contributions to transpiration of the various soil layers were greatly affected by the rainfall pattern. The MP is a shallow-rooted species with root density decreasing sharply below 1.0 m (Jiang et al., 2002; Zhu et al., 2005; Zhu et al., 2008). This indicates the soil moisture in the upper 1.0 m provides the principle water source for transpiration (Su et al., 2006; Wei et al., 2013; Song et al., 2014). In our study, the rapid recovery of $T_s$ as well as $\overline{\theta}_{0-1.0\,m}$ following rain (Figs. 3 and 6) shows that MP is very sensitive to changes in the available water in this soil layer. Uptake by the shallow roots   decreased very

significant as this soil layer dried out (Fig. 6). The fine roots of Scots pine die quickly under drought conditions (Vanguelova and Kennedy, 2007). Therefore, fine root death may explain why following rainfall, post-stress sap flux recovery is only partial

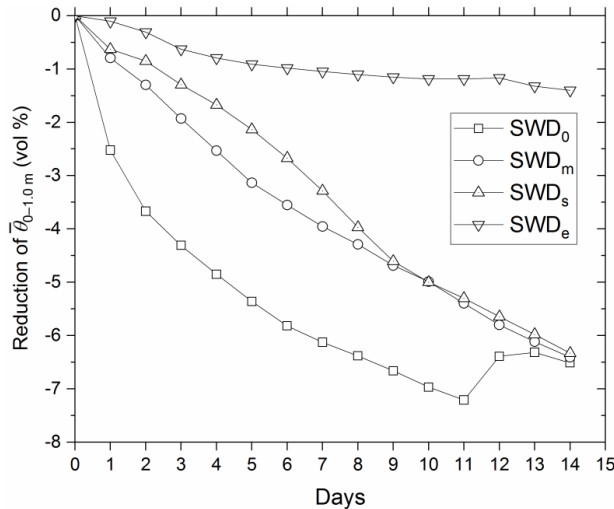

in dry years in our study. Here, the stand transpiration of MP fell quickly with REW < 0.24. We infer that this indicates the soil moisture threshold for our site. This is similar to thresholds reported for Scots pine REW < 0.25 (Vincke and Thiry, 2008) and REW < 0.20 (Lagergren and Lindroth, 2002).

**Figure 7** There will be an effective rain event every 14 days averagely in our site (Rainfall intensity is more than 10 mm per times), which acted as a window to analyze the decrease of soil water during this period. Scatter-line plot described the relationships between decrease of upper soil moisture ($\overline{\theta}_{0-1.0\,m}$) and time under different degree of drought levels which were defined in **Table 2**.





### 4.3 Groundwater as a source of water for transpiration

Mongolian Scots pine also is a dimorphic-rooted species where, the maximum taproot depth in a sandy soil can up to 2.7 m (Canadell et al., 1996), and even to 5.2 m for a 42-year-old tree in a sandy soil near our site (Jiang et al., 2002). This possibly enables them to switch from the upper to the deep water source (i.e. groundwater) depending on water availability (Barbeta et al., 2015; Hentschel et al., 2016). This is particularly likely to occur when soil moisture in the upper soil layers (0–60 cm) declines to about 3.6 % (Wei et al., 2013; Song et al., 2016). In our site, $g_w$ lowered from 5.03 ± 0.14 m (in 2013), to 5.13 ±

0.07 m (in 2014) to 5.47 ± 0.09 m (in 2015). From late 2014, the value of $g_w$ was always far deeper than 5.2 m (**Fig. 3e**) and thus unlikely accessible directly by our instrumented trees if their tap roots were shallower than 5.2 m. However, in the extreme drought (SWD$_e$, with minimum $\overline{\theta}_{0-1.0\,m}$ as low as 2.3 %), we recorded a clear diurnal pattern of sap flux with the much reduced daily $T_s$ (mean 0.63 mm day$^{-1}$, or 34.6 % of that for SWD$_0$). Hence, we infer that significant groundwater contributions to $T_s$ occurred only under extreme drought conditions though determining just what proportion of that water came from the ground-

water or from tree storage is beyond the scope of this study. It has been reported that as rainfall decreases, tree dependence on groundwater increases (Kume et al., 2007).

### 4.4 Water transport resistance due to environmental factors

   There is a complex interplay between the various meteorological factors - solar radiation, vapour pressure deficit, air temperature, windspeed and relative humidity - ultimately these are all driven by the Earth's rotation. Directly or indirectly, each of

these influences transpiration in a tree and thus sap flux. Here, these variables were aggregated into the variable ET$_0$, which serves as a good index of atmospheric water demand (Zha et al., 2010). Therefore, as expected, changes in ET$_0$ trigger a prompt plant response in terms of transpiration. Meanwhile, changes in some other meteorological variables, such as precipitation (and hence soil water), have their effects on transpiration but over a much longer temporal scale (Yan et al., 2016). Our analyses show that MP shows a strong reduction in ratio $T_s / ET_0$ as a drought intensifies (Figs. 4 and 8a). This behavior has also been

found for Scots pine in Europe (Poyatos et al., 2005) and representing the strong effects of stomatal regulation for controlling the rate of water loss. Meanwhile, stepwise regression analyses show that the relationship between the ratio $T_s / ET_0$ and the three variables: $g_w$, REW and $D$, was $T_s / ET_0 = 1.567 - 0.242\ g_w + 0.231\ REW - 0.092\ D$ ($R^2 = 0.49$, $n = 552$, $P < 0.001$). Clearly, $g_w$ played an important role in controlling water fluxes over the three years of our trial, the significant fall in $g_w$ seems to explain the only partial recovery of $T_s$ following heavy rain during the later severe periods of drought. We calculate that the

ratio $T_s / ET_0$ decreased at a rate of 0.35 per 1.0 m decline of groundwater level (**Fig.8b**).

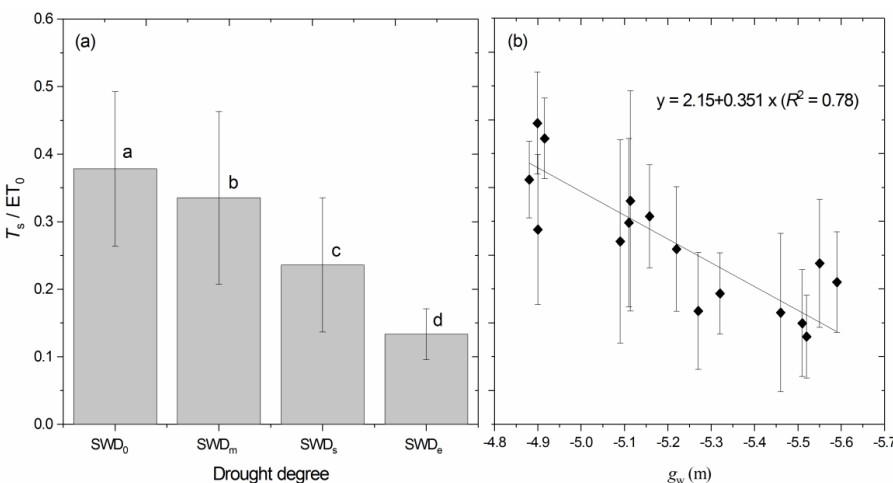

**Figure 8** Quotient of daily sap flux and reference evapotranspiration ($T_s / ET_0$) underwent an almost stepwise decline with drought in (a) and varied with the level of the groundwater table ($g_w$) in (b). The degree of drought is defined in **Table 2**. Values of $T_s / ET_0$ followed by different letters are significantly different at $P < 0.05$ by two-factor univariate ANOVA (post hoc Tukey HSD).





### 4.5 Water budget at stand scale

In the MP in this study, the reduction in stand transpiration due to soil water shortage was 37–40 % (**Table. 3**). This compares with reported values of 40 to 80 % with different species and in different habitats (Leuzinger et al., 2005; Gartner et al., 2009; Betsch et al., 2011). During our three-year study, average cumulative $T_s$ values during a whole growing season were in the

range 175–290 mm. This is higher than 74–88 mm in a sparse forest of 150-year-old Scots pine growing in an inner Alpine dry valley (Wieser et al., 2014), 82–113 mm in northeastern Germany (Lüttschwager et al., 1998) due to having different leaf area indices (LAI values were 1.47–1.61 in our site), different climates and soil conditions. To calculate the hydrological budget, we adopted the bulk rainfall interception rate by the canopy and understorey monitored in similar stand; this totaled about 23 % of total rainfall (Jiao, 1989; Jiao, 2001; Han et al., 2015). The direct drainage of rainfall into the groundwater was

assumed negligible (this is usually less than 1 % of annual rainfall - data not shown). As a result, the water budget of the plantation is generally very tight indeed with net rainfall over two or three months each year being well below the value of $T_s$ (**Table. 3**). Hence, the soil fails to recharge at all well in most seasons. In particular, the minimum plant-available water storage from the upper soil profile (0–1.0 m) ($\theta_{c\text{-min}}$) in the July period of the two dry years was significantly less than $T_s$. Hence, severe depletion of the shallow soil water reserves will have occurred along with some depletion of the deeper groundwater in order

to meet the transpiration demands of the trees. It is reasonable to infer that, about 8.7 % of total sap flux in July 2014 and 34.6 % in July 2015 came from these deeper soil water resources, most likely from the groundwater.

**Table 3** Total stand water use ($T_s$), net rainfall ($R_{net}$) and the minimum plant-available soil water storage in the upper 1.0 m soil profile ($\theta_{c\text{-min}}$) during the growing seasons over the whole trial period of three years. This period includes the wet year

2013, the dry year 2014 and the second dry year 2015.

| Month | year 2013, mm | | | year 2014, mm | | | year 2015, mm | | |
|---|---|---|---|---|---|---|---|---|---|
| | $T_s$ | $\theta_{c\text{-min}}$ | $R_{net}$ | $T_s$ | $\theta_{c\text{-min}}$ | $R_{net}$ | $T_s$ | $\theta_{c\text{-min}}$ | $R_{net}$ |
| May | 54.0 | 81.0 | 17.4 | 37.3 | 66.4 | 48.7 | 33.2 | 36.8 | 53.6 |
| Jun. | 51.8 | 61.0 | 41.0 | 36.1 | 57.8 | 97.9 | 33.7 | 47.4 | 74.1 |
| Jul. | 61.3 | 92.9 | 195.3 | 37.0 | 33.7 | 37.1 | 28.0 | 18.3 | 27.1 |
| Aug. | 57.9 | 68.8 | 72.4 | 27.8 | 42.2 | 70.1 | 23.9 | 41.4 | 92.9 |
| Sep. | 41.7 | 65.3 | 19.7 | 29.9 | 39.2 | 12.3 | 32.5 | 63.3 | 24.9 |
| Oct. | 23.5 | 63.5 | 55.6 | 14.4 | 23.5 | 9.4 | 22.3 | 50.8 | 16.8 |
| Sum | 290.2 | – | 401.4 | 182.5 | – | 275.5 | 173.6 | – | 289.4 |
| $T_s$ / $R_{net}$ | 0.72 | – | – | 0.66 | – | – | 0.60 | – | – |

### 4.6 Strategies adapted to site-related rainfall pattern

Transpiration in a coniferous forest is often conservative with relatively low values of canopy conductance (Levitt, 1980). For
instance, Scots pine has a rather conservative water use strategy with a very plastic response to intermittent dry periods with high use of stored water (Arneth et al., 2006; Verbeeck et al., 2007). In our study, we find MP is more moderate in its water consumption than many broad-leaved forest tree species growing nearby (e.g. *Populus* spp) (Zhu et al., 2005). Therefore, it contributes less to the groundwater table decline (0.1 m per year)(Song et al., 2016) than the more extensive and/or intensive agricultural land uses. The lateral roots of an MP tree can extend laterally to about 0.65-times tree height (Jiang et al., 2002;
Su et al., 2006). This helps it obtain water from the upper soil layers efficiently (Song et al., 2014). The ability of MP to maintain low levels of sap flux even during extreme drought, indicates its ability to access the water large volumes of soil (Waromg et al., 1979). Recharge of soil water following rainfall is quickly reflected in increased sap flux, which reflects the great resilience of MP to drought. This may explain why this stand managed to survive the eight-year dry period from 1996 to 2004 (**Fig. 1**).

## 5 Conclusions





The relationships between sap flux and atmospheric demand, soil moisture and groundwater table were analysed to show to what extent and how the water use of MP in sandy soil is limited by drought. We found that MP were relatively conservative in their water usage, never exceeding 3.03 mm day$^{-1}$ in 2013, 1.75 mm day$^{-1}$ in 2014 and 1.59 mm day$^{-1}$ in 2015. Stand transpiration amounts during the growing season were 290 mm (year 2013), 182 mm (year 2014) and 175 mm (year 2015),

which accounted for about 46–56 % of total rainfall. In general, the sap flux in MP responded strongly to atmospheric demand under well-wetted soil conditions but the response weakened as the intensity of drought increased. The mean daily sap flux, reduced with drought by as much as 65 % as the duration and intensity of drought increased, but recovered fully after relief of a moderate drought by heavy rainfall in 2013. However, recovery was only partial (57–58 %) after relief of a severe drought in 2014 and of an extreme drought in 2015. The magnitude of the reduction in sap flux under the severe and extreme droughts

in 2014 and 2015 ($\overline{\theta}_{0-1.0\,m}$ < 6.0 %, REW < 0.24) indicates the soil moisture threshold which limits water extraction by MP trees in these sandy soils. Our results suggest that the degradation in this MP plantation is attributable to the combined effects of (1) the large intra-seasonal and inter-annual variances in rainfall in this region, (2) the sandy soil and (3) a progressive decline in the level of the groundwater table.

**Acknowledgements**

This work was supported by the Fundamental Research Funds for the Central Non-profit Research Institution of CAF (No.CAFYBB2014MA013), the Major State Basic Research Development Program of China (973 Program) (No. 2013CB429901) and the National Natural Science Foundation of China (No.31570704). We thank Dr. Alexander Lang for his useful discussions and suggestions. Field support for this research was provided by Zhanggutai National Studies Station for Desert Ecosystem of China (CDERN).

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
