# Peer review of "Rainfall pattern greatly affects water use by Mongolian Scots pine on a sandy soil, in a semi-arid climate"

_Biogeosciences, 2017_

## Referee Comment (RC1) · Anonymous Referee #1 · 20 Mar 2017

Identifying the water use by Mongolian Scots pine (Pinus sylvestris var. mongolica) is always the core of ecological restoration in arid and semiarid regions. In this study, the authors have analyzed the sap flux density, transpiration and its relationship with precipitation and soil moisture. The manuscript is suitable to publication in this journal; however, the author should make a minor revision mainly because of the poor organization of results and discussion part. My comments are shown as follows: Title: Please give the full Latin name of Mongolian Scots pine. Abstract: The abstract is too long, please shorten it and try to use more concise and simple sentences. Line 15-25: the sentences were too long, the writing should be shorter and clearer. Line 23: how long did the daily Ts recovered completely? Line 25: did you have any results of

trees growth influenced by large intra- and inter-annual variances in rainfall? Did you analyze the rainfall intra- and inter-annual variances? Line 27: What does MP mean? Introduction: Line 34: I think here "However" is better than "nevertheless". Line 36: timings here means rainfall interval? Line 47-48: rewrite this sentence. Line 60-63: this paragraph was mainly talked about soil water availability, thus, I think those sentences were not suitable here. Line 67-69: did this hypothesis reasonable? The author declared that 85% of the roots were in the shallow soil layers, but there were still 15% of the roots in the deep layers, they may absorb water from the groundwater, and in addition, what is the groundwater table in this region? Line 70-73: I think it should be removed to the method section. Line 74-76: please rewrite this sentence. 2 Materials and methods: please give a figure of your study site. Line 90: I suggest giving the rainfall amount (average and the year of 2013, 2014 and 2015) in Figure 1. Line 101: what types of sensor? 3 Results Line 199-203: the sentence was too long, I think a table here would be better. Line 244-245: remove this sentence to the method section. Line 246-249: rewrite this sentence. Line 260: how many rainfall events in the dry-wet shift in 49 days? Especially before the one off big rain event, is there any other small rainfall events? And in my opinion, I think after the rain, especially the number of rewetting days should be same if you want to compare in different years. 4 Discussion Line 277-283: remove this sentence to the results section. Line 304-306: delete or remove to the result section. 4.5 I think this part should be removed to the result section.

---

## Referee Comment (RC2) · Anonymous Referee #2 · 25 Mar 2017

The MS by the authors presents data from 3 years of measurements of Mongolian Scots Pine. While such data will be very useful for our understanding of this species plant water use and strategies, as well as ecosystem functioning in which this species is found, the collection of sap flow data and subsequent statistical analyses were poor and do not warrant publication.

The authors deployed the Granier technique, or thermal dissipation probe (TDP), to measure sap flow in Mongolian Scot's Pine. The TDP sensor only has one measurement point that the authors installed at an unknown depth in the sapwood. The width, or depth, of the sapwood is also unknown. From the single measurement point, sap flow data were scaled upwards to tree and then entire forest stand. Such a scaling

approach no doubt introduced significant errors as it is known, from a vast amount of research, that sap flow varies across the radial profile of sapwood. Measuring at a single point in the sapwood will then significantly over- or under-estimate true sap flow. Furthermore, the Granier technique is an empirical technique and requires a species specific calibration. The authors chose a generic calibration which may lead to further errors in their estimate of sap flow. Therefore, the sap flow data collected by the authors cannot reliably be used to estimate total tree or stand water use.

The statistical approach by the authors is also incorrect. The authors analysed a series of collinear variables across a series of univariate or a multivariate regression analysis. All the environmental variables are related to each other therefore the author's approach will introduce a significant amount of error. Collinearity in regression models is a commonly overlooked statistical error. I recommend the author's read the chapter on collinearity in Quinn and Keough (2002) "Experimental Design and Data Analysis for Biologists". The authors can run their predictor variables through a Principle Components Analysis to reduce their related variables to a series of unrelated variables. The factor scores that come out of the PCA can then be used in a multiple linear regression analysis as the predictor variables.

The analyses and results have the feeling that the authors are fishing through their data set looking for significant patterns and then presenting those patterns. There is no systematic analytical approach and it is extremely difficult to follow and comprehend. For example, several drought periods are declared based on REW and a regression analysis between Ts and ETo is presented in Figure 4. But what is the significance of REW <= 0.24? Why is it not 0.25 or 0.23? These REW values may be important but it just seems some random numbers were chosen.

The MS would be far stronger, and much easier to read and comprehend, if there was a systematic and biologically realistic designation of drought periods.

Figure 5 also suffers from this problem. What is EW? Is this something the authors

have just created? If it is so important, why is it being introduced deep into the Results section and not in the Introduction section? Why is the power model important? Does it have a slightly better R2 than a linear regression model? But does it carry greater explanatory power in terms of an AIC analysis or other statistical approach?

Throughout the MS, there are several grammatical and spelling errors. For example, "sapflow" should be "sap flow".

Other points:

- the linear regression in figure 1 is in appropriate and statistically incorrect. Is the purpose of the regression to demonstrate that rainfall decreased and temperature increased over this time period? If yes, probably consult the climate science literature to determine the methods they use to statistically quantify such changes.

- can you please provide more details about the soil profile – for example, is the texture consistent down the profile or are there distinct horizons? Is the texture described in the text only for the top 40cm of soil where, presumably, most of the MP root activity is?

- where were the soil moisture sensors installed? Next to the weather station outside of the plot? or somewhere inside of the plot?

- what was the sapwood width of MP?

- why were the TDP sensors installed on the north-face of the trees? Presumably, this is where sap flow would be lowest around the circumference of the trunk.

---

## Referee Comment (RC3) · Anonymous Referee #3 · 27 Mar 2017

Drought is the key limiting factor on afforestation in the arid and semiarid areas, also is the main reason for degradation of the artificial forests. The study presented a theme of great interest: the response of sap flux density of Mongolian Scots pine (Pinus sylvestris var. mongolica) to increasing levels of drought from wet to moderate-drought, severe-drought and extreme-drought, as well as its relationship with variable precipitation, soil moisture and groundwater table. It is very useful with regard to explaining the degradation of Mongolian Scots pine forests, and guiding afforstation and management of Mongolian Scots pine forests. The MS is suitable to publication in this journal. However, some small problems need to be modified in the present version of the manuscript. My comments are shown as follows: 1. In abstract, what is MP

on Line 27? It should be listed in the beginning sentence. 2. In order to understand the sandy soil conditions, suggest author to provide more details, including texture and nutrient content about the soil at the research area at the part 2.1 Site description. 3. In conclusion, authors considered that the degradation in this MP plantation was attributable to the combined effects of rainfall, the sandy soil and groundwater table. So some information on growth status of Mongolian Scots pine in research area should be provided at the part 2.2 Trial plots and sample trees. Degradation or not? 4. How many sets of soil moisture sensors were installed? Where the soil moisture sensors were installed? Inside or outside of the plot? How far were they from plant stand? 5. In Fig.4, Fig.5 and Fig.8, regression equations should be carried out significance test, and significant coefficient should follow the linear regression equations in figures. 6. The driving factors for the degradation of artificial forests not only include drought, pest and fire disaster, but also include man-made factors, especially high afforestation density. In discuss part, the author should discuss the planting density can be loaded by variable rainfall in research area. Whether the degradation of MP was related to the planting density? And can you put forward or suggest the rational plant density of MP on the sandy soil in the different areas with different rainfall? 7. Throughout the MS, there are several spelling errors. For example,"sapflow" on Line 101 should be "sap flow", "annualprecipitation" on Line 87 should be "annual precipitation". 8. Precipitation and rainfall were used in MS, including Figure. However, some scholars consider the Precipitation and rainfall are different. Precipitation should be in any form, covers all forms of water being released by the atmosphere, include rainfall, snow, drizzle, sleet. 9. In conclusion, first sentence "The relationships between sap flux and atmospheric demand, soil and groundwater table were analysed to show to what extent and how the water use of MP in sandy soil is limited by drought."should be deleted or removed.

---

## Author Comment (AC1) · 19 Apr 2017

General comments: Identifying the water use by Mongolian Scots pine (Pinus sylvestris var. mongolica) is always the core of ecological restoration in arid and semiarid regions. In this study, the authors have analyzed the sap flux density, transpiration and its relationship with precipitation and soil moisture. The manuscript is suitable to publication in this journal; however, the author should make a minor revision mainly because of the poor organization of results and discussion part. Response: We thank referee for the helpful comments. After discussion with co-authors, we thoroughly revised the manuscript and listed in supplement. Specific comments: Title: Please give the full Latin name of Mongolian Scots pine. Response: We added the

full Latin name, that is: Mongolian Scots pine (Pinus sylvestris var. mongolica), and the title revised as "Severe drought greatly reduces sap flow of Mongolian Scots pine (Pinus sylvestris var. mongolica) and its recovery ability in a sandy and semi-arid environment". Abstract: The abstract is too long, please shorten it and try to use more concise and simple sentences. Response: We revised in new manuscript. Line 15-25: the sentences were too long, the writing should be shorter and clearer. Response: These long sentences were rewritten in new manuscript. Line 23: how long did the daily Ts recovered completely? Response: In our revised designation of drought period (Table 2), there was no drought in 2013, so there is no recovery following heavy rainfall in 2013. Line 25: did you have any results of trees growth influenced by large intra- and inter-annual variances in rainfall? Response: We did not measure the variable about growth and we replaced it with the word "water use". Did you analyze the rainfall intra- and inter-annual variances? Response: The intra- and inter-annual variances were analyzed in the first two paragraphs of section 3.1 Line 27: What does MP mean? Response: We replaced it with the full name, that is Mongolian Scots pine. Introduction: Line 34: I think here "However" is better than "nevertheless". Response: We replaced the "nevertheless" with "However" in Line 30 of revised manuscript. Line 36: timings here means rainfall interval? Response: Yes, we revised it to "interval" in Line 32. Line 47-48: rewrite this sentence. Response: We rewrote it as "It is found in the Daxinganling Mountains and in Honghuaerji on the Hulun Buir sandy plains of the northeast (Zhu et al., 2008; Zheng et al., 2012)" in Line 41-43 of revised manuscript. Line 60-63:this paragraph was mainly talked about soil water availability, thus, I think those sentences were not suitable here. Response: We realized the incorrectness of those sentences, we deleted it in revised manuscript. Line 67-69: did this hypothesis reasonable? The author declared that 85% of the roots were in the shallow soil layers, but there were still 15% of the roots in the deep layers, they may absorb water from the groundwater, and in addition, what is the groundwater table in this region? Response: The MP is a shallow-rooted species with over 85 % of roots located in the upper 0.4 m of the soil profile and sharply decreased root density with depth down to 1.0 m in our

site (Su et al., 2006). So it is reasonable to hypothesize that the most water absorbed by Mongolian Scots pine is from the upper 1.0 soil layer. However, we also think that the taproot of some big trees could absorb deeper soil moisture even groundwater directly. The groundwater table in our site during the study period decreased from 5.6 m to 4.8 m, indicating a existence of deep root. We discussed it in section 4.2 in revised manuscript. Line 70-73: I think it should be removed to the method section. Response: We removed it to the section 2 Materials and methods, and describe it respectively. Line 74-76: please rewrite this sentence. Response: These sentences have rewritten in Line 62-67 of revised manuscript. 2 Materials and methods: please give a figure of your study site. Response: We added the location map of study area as Fig.1 Line 90: I suggest giving the rainfall amount (average and the year of 2013, 2014 and 2015) in Figure 1. Response: We changed the figure into anomalies of air temperature and precipitation (%) in our revised manuscript (Fig.2), the rainfall amount of historical means is provided in Line 78 and these measurements years in Line 189 of the revised manuscript. Line 101: what types of sensor? Response: The type of sensor is Granier-type thermal dissipation method (Dynamax Inc., Houston. TX. USA) with 3-cm long probe. We clarified in Line 153-154 in the revised text. 3 Results Line 199-203: the sentence was too long, I think a table here would be better. Response: The sentence was rewritten in Line 189-191 of revised manuscript. Line 244-245: remove this sentence to the method section. Response: We deleted it in revised manuscript. Line 246-249: rewrite this sentence. Response: We deleted it in revised manuscript. Line 260: how many rainfall events in the dry-wet shift in 49 days? Especially before the one of big rain event, is there any other small rainfall events? And in my opinion, I think after the rain, especially the number of rewetting days should be same if you want to compare in different years. Response: In our site, the rainfall intensity lower than 10 mm per day is unavailable for trees due to canopy interception and soil evaporation. There were some small rainfall events during this period (Fig. 4). After a heavy rain, for example, 47.6 mm per day in 2014 and 82.4 mm per day in 2015, the soil was well recharged and the relative extractable water

(REW) increased quickly. Yes, after this heavy rain, the number of rewetting days and plant recovery days could be the same between years. But the recovery ability (maximum Ts) differed due to the degree and duration of the drought. 4 Discussion Line 277-283: remove this sentence to the results section. Response: We think what contained in these sentence are indirect information that help explain the quick decline of soil moisture in sandy soil in our site. We re-organized it in section 4.1 in revised manuscript, because we want only focus on the transpiration of trees in main text and we used this only to support our discussions. Line 304-306: delete or remove to the result section. Response: We re- organized it in section 4.3 in revised manuscript. 4.5 I think this part should be removed to the result section. Response: We deleted this section in our revised manuscript.

Please also note the supplement to this comment:
http://www.biogeosciences-discuss.net/bg-2017-69/bg-2017-69-AC1-supplement.pdf

---

## Author Comment (AC2) · 19 Apr 2017

General comments: Drought is the key limiting factor on afforestation in the arid and semiarid areas, also is the main reason for degradation of the artificial forests. The study presented a theme of great interest: the response of sap flux density of Mongolian Scots pine (Pinus sylvestris var. mongolica) to increasing levels of drought from wet to moderate-drought, severe-drought and extreme-drought, as well as its relationship with variable precipitation, soil moisture and groundwater table. It is very useful with regard to explaining the degradation of Mongolian Scots pine forests, and guiding afforstation and management of Mongolian Scots pine forests. The MS is suitable to publication in this journal. However, some small problems need to be

modified in the present version of the manuscript. Response: We thank referee and greatly appreciate the thoughtful and constructive comments. We have fully considered the referee's comments in the revision and improved the manuscript listed as supplement. Special comments: 1. In abstract, what is MP on Line 27? It should be listed in the beginning sentence. Response: MP means Mongolian Scots pine, we replaced it with the full name in revised text. 2. In order to understand the sandy soil conditions, suggest author to provide more details, including texture and nutrient content about the soil at the research area at the part 2.1 Site description. Response: We added related information about soil profile in Line 82-85 in revised manuscript, that is: The soil is sandy with a sedimentary aeolian sand layer more than 3 m and an ancient alluvial sand layer with the total depth more than 126 m (Jiao, 1989). The mean bulk density of the upper 2 m soil layer is 1.61 g cm-3. The mean soil texture is 83 % of sand (> 0.05 mm), 9 % of silt (0.05–0.002 mm) and 8 % of clay (< 0.002 mm). The organic matter content is 0.3–1.0 g kg-1. 3. In conclusion, authors considered that the degradation in this MP plantation was attributable to the combined effects of rainfall, the sandy soil and groundwater table. So some information on growth status of Mongolian Scots pine in research area should be provided at the part 2.2 Trial plots and sample trees. Degradation or not? Response: We added the growth related information in Line 91-92 of revised manuscript, that is : The growth of trees in the experiment was normal in 2013, however, the leaves of trees in 2015 turned to grey slightly. The obvious defoliation or death did not occurred in 2015. There was no obvious degradation discerned in our short measurement period (three years). 4. How many sets of soil moisture sensors were installed? Where the soil moisture sensors were installed? Inside or outside of the plot? How far were they from plant stand? Response: We added the soil moisture sensors related information in Line 136-138 of revised manuscript, that is: Three placements in experiment area were measured. Each placement was set between four neighborhood sample trees. 5. In Fig.4, Fig.5 and Fig.8, regression equations should be carried out significance test, and significant coefficient should follow the linear regression equations in figures.

[Figure]

Response: We added the significance in all statistical figures in revised manuscript. 6. The driving factors for the degradation of artificial forests not only include drought, pest and fire disaster, but also include man-made factors, especially high afforestation density. In discuss part, the author should discuss the planting density can be loaded by variable rainfall in research area. Whether the degradation of MP was related to the planting density? And can you put forward or suggest the rational plant density of MP on the sandy soil in the different areas with different rainfall? Response: We added the planting density related discussion in section 4.4, Line 295-298 of revised manuscript. We agreed that the degradation of Mongolian Scots pine was related to the stand density. 7. Throughout the MS, there are several spelling errors. For example,"sapflow" on Line 101 should be "sapflow", "annualprecipitation" on Line 87 should be "annual precipitation". Response: We check the spelling errors throughout full text. 8. Precipitation and rainfall were used in MS, including Figure. However, some scholars consider the Precipitation and rainfall are different. Precipitation should be in any form, covers all forms of water being released by the atmosphere, include rainfall, snow, drizzle, sleet. Response: We use precipitation to indicated total water capture in the text as a general term, but use rainfall when refer to the water capture by rain event during the growing seasons. 9. In conclusion, first sentence "The relationships between sap flux and atmospheric demand, soil and groundwater table were analysed to show to what extent and how the water use of MP in sandy soil is limited by drought."should be deleted or removed. Response: We realized the incorrectness and deleted it.

Please also note the supplement to this comment:
http://www.biogeosciences-discuss.net/bg-2017-69/bg-2017-69-AC2-supplement.pdf

[Figure]

**Supplement:**

**Severe drought greatly reduces sap flow of Mongolian Scots pine (*Pinus sylvestris* var. *mongolica*) and its recovery ability in a sandy and semi-arid environment**

HongZhong DANG[1,a], LiZhen ZHANG[2], WenBin YANG[1], JinChao FENG[1], Hui HAN[3], Wei LI[1]

[1] Institute of Desertification Studies, Chinese Academy of Forestry, Beijing, 100091, China

[2] Institute of Resources and Environment, China Agricultural University, Beijing, 100094, China

[3] Institute of Sand Fixation and Afforestation of Liaoning Province, Fuxin, 123000, China

[a] *Correspondence to*: HongZhong DANG (hzdang@caf.ac.cn)

**Abstract.** Trees growing in water limited ecosystems are often exposed to the significant challenges of soil water stress due to low precipitation and high variation. In this study, we aimed to quantify the water use of Mongolian Scots pine (*Pinus sylvestris* var. *mongolica*) growing on a sandy soil, in a region characterised by an erratic rainfall pattern. Measurements were made over three successive years of contrasting annual rainfall - a wet year (2013), a dry year (2014), and a second dry year (2015). Over the three years, sap flux density ($J_s$) was measured at individual tree level for 25 tree samples, and were up-scaled to estimate tree transpiration at plot level ($T_s$). Due to the high variation of rainfall in three years, the measurements reflected the tree response to wide range of water stress from wet (2013), mild-drought (2014) to severe-drought (2015). The daily transpiration $T_s$ of trees during growing seasons was 3.7 mm day$^{-1}$ in 2013, 2.6 mm day$^{-1}$ in 2014, but sharply decreased to 1.4 mm day$^{-1}$ in 2015 after a long-period drought stress, resulting a large difference in total annual stand transpiration as 357 mm in 2013, 268 mm in 2014 and 149 mm in 2015. Under a long-period of drought stress in late season in 2014 and early season in 2015, the recovery of $T_s$ was incomplete (63–69%). The erratic rainfall and sandy soil coupling with a declining groundwater table, tree water use fluctuated widely over quite short time scales (months or weeks). Our results help elucidate the interplay between the effects of the atmosphere and soil moisture on tree water use, and highlight the negative effects of drought on water use of mature forest tree. Our findings provide the evidence for the observed premature degradation of these Mongolian

Scots pine plantations in terms of an eco-hydrological perspective.

**Keywords:** sap flux; Mongolian Scots pine (*Pinus sylvestris* var*. mongolica* Litv*)*; soil water availability; water stress; sandy soils; semi-arid climate.

**1 Introduction**

Reforestation has been used widely in semi-arid areas to control soil erosion, to capture carbon and to serve as wind breaks (D'Odorico and Porporato, 2006). However, trees growing in severely water-limited ecosystems are often exposed to significant challenge due to insufficient soil water (Wesche et al., 2011; Su et al., 2014). Many factors influence the amount of soil water and its availability to vegetation, for instance, the amounts and timings of rainfall interval, soil water capacity, root water-uptake capacity and the availability of alternative water sources such as groundwater (Meinzer et al., 2006). Under climate change, the increasing in the frequency and severity of drought decreases soil water availability in the future (Leo et al., 2013). This would increase tree mortality rate through excessive competition for water and thus influences the structure and function of forest ecosystems (Barbeta et al., 2015). Quantification of water use by trees at individual and forest levels could help us to understand how environmental factors affect their water usage. It is necessary to properly assess the impacts of climate change on ecological and hydrological processes in these fragile ecosystems (Bovard et al., 2005). These knowledge would allow us to make better forest establishment decisions and management actions.

Mongolian Scots pine (*Pinus sylvestris* var. *mongolica*, MP), a geographical variety of Scots pine (*P. sylvestris*), is naturally widely distributed in northern China and in parts of Russia and Mongolia. It is found in the Daxinganling Mountains (50 °10 ′–53°33′ N, 121°11′–127°10′ E) and in Honghuaerji on the Hulun Buir sandy plains of the northeast (Zhu et al., 2008; Zheng et al., 2012). The MP is a popular species for reforestation in northern China due to its traits of good drought and cold resistance. Consequently, more than $6.7 \times 10^5$ ha of MP plantations have been established to control desertification, in the great project of the Three-North Shelter Forest Program (TNSFP) launched in China from 1978 (Zheng et al., 2012). Unfortunately, serious degradation and considerable concern has occurred in these plantations since the mid-1990s, such as poor tree health and numerous tree death, particularly on the sandy soils in southern Horqin (42°43′ N, 122°22′ E, our study area) (Jiao, 2001; Zhu et al., 2008) (**Fig. 1**).

[Figure]

**Figure 1** Location and eviornment of Horqin region in Liaoning province, China (study area)

A key driver for the degradation in water-limited ecosystems is regional low and erratic precipitation, which reduces soil water availability (Mereu et al., 2009). In semi-arid southern Horqin, three main soil-water related factors causes the degradation, i.e. the high inter-annual variation in precipitation, the high intra-seasonal variability in precipitation and the declining groundwater table. The forest's sensitivity to drought is highly species-specific, climate-specific and site-specific. However, it remains unclear how, and to what extent, these three factors are responsible for the degradation recorded in this region.

In general, plants native to arid and semi-arid environments have developed a wide range of water-use strategies to cope with drought, e.g. by 'water saver' plants to avoid drought by minimizing transpiration with limiting leaf area growth, or by defoliation and stomatal closure (Levitt, 1980; Gartner et al., 2009; Chirino et al., 2011). The MP is a shallow-rooted species and more than 85 % of roots system grows in the upper 0.4 m soil layer. The root density is sharply decreased below 1.0 m soil depth (Su et al., 2006).

We hypothesize that on the long-period drought in semi-arid sandy environment raising from the high variation of pre- cipitation and low water holding capacity is the major reason causing the degradation of forest. The aims of this study were:

(1) to quantify daily water use of MP based on sap-flux measurements in relation to the three contrasting precipitation years; (2) to determine the relationships between $J_s$ and the main meteorological variables and soil water availability over the ex- tended (three-year) period; (3) to explore the effect of the severity of the drought stress over number of cycles of soil wetting and drying on daily water use and recovery ability.

[Figure]

**Figure 2** Annual variations of precipitation (a), mean air temperature (b) at Zhanggutai. Grey color indicates the data before the experiment (1983-2012) and black for the years of experiments (2013-2015). The dashed lines indicate the linear regressions over the whole period.

**2    Materials and methods**

**2.1    Site description**

The trial was carried out at the Zhanggutai National Desertification Control Trial Station located at the southern edge of the

Horqin region in Liaoning province, China (122° 22′ E, 42° 43′ N, at 226.5 m a.s.l.) (**Fig. 1**). The experiment was conduced in a 40 ha plantation with 35-year old MP. Tree density was 625 trees per ha. Management interventions and other human dis- turbances were limited by the installation of a secure fence around the experiment field. The site has a semi-arid, continental climate with a mean annual temperature of 7.9 °C, a frost-free period of 150–160 days per year, a mean annual evaporation of

1553 mm and a long-term annual mean precipitation of 475 mm ($P_{ave}$) over the last 30 years (1983–2012) with coefficient of variance of 0.27 (Zhu et al., 2005). Over the long period, there have been a number of consecutive dry years. For instance, annual precipitation between 1996 and 2004 were below $P_{ave}$ (**Fig. 2a**). Usually, about 60 to 70 % of annual rainfall occurs in the three months from June to August. The value of annual temperature over the last 30 years was increased slightly at a rate of 0.03 °C yr$^{-1}$ (**Fig. 2b**), while annual precipitation was slightly decreased with a rate of 2.0 mm yr$^{-1}$ (**Fig. 2a**). The soil is sandy with a sedimentary aeolian sand layer more than 3 m and an ancient alluvial sand layer with the total depth more than

126 m (Jiao, 1989). The mean bulk density of the upper 2 m soil layer is 1.61 g cm$^{-3}$. The mean soil texture is 83 % of sand (>

0.05 mm), 9 % of silt (0.05–0.002 mm) and 8 % of clay (< 0.002 mm). The organic matter content is 0.3–1.0 g kg$^{-1}$. The understory plant species in the forestry are *Acer pictum* subsp. *mono* Maxim, *Crataegus pinnatifida var. major N. E. Brown.*,

*Lespedeza bicolor* Turcz., *Artemisia halodendron* Turcz et Bess., *Cleistogenes chinensis* Maxim.

**2.2 Experimental design and samplings**

To break the prevailing northerly winds, the MP were planted in a square-grid pattern with 4 m for both row spacing and plant distance. Total area of experiment was 400 m$^2$ (20×20 m) containing 25 trees. All trees in the area were planted at same year in sole system surrounded by a wire fence (**Fig. 3**). The growth of trees in the experiment was normal in 2013, however, the leaves of trees in 2015 turned to grey slightly. The obvious defoliation or death did not occurred in 2015. Sap flow sensors were installed in each tree (totally 25 trees) in experimental area in 2013. Due to the damage of sensors, 22 left in 2014 and 13

left in 2015. The characteristics of the sampled trees are shown in **Table 1**. Diameters at breast height (DBH) were measured with a diameter tape and tree height with an altimeter. The thickness of bark, sapwood and heartwood were measured by sampling core with a Pressler increment borer at breast height. Thickness measurements were made with a Vernier caliper with tissue boundaries identified based on color. In our Mongolian Scots pine, the sapwood colour was yellow-white and that of the heartwood was tan. A few drops of methyl orange solution helped define the interface where the boundary was indistinct. The

DBH, tree height and sapwood areas of the sampled trees in 2013, 2014 and 2015 were not significantly different ($P > 0.05$), indicating a good uniformity of testing trees.

[Figure]

**Figure 3** Sketch map of 25 sample trees (stars) planted in a 4×4 m spaced square grid of about 400 m$^2$ (dashed line is border fence). Tree ages were identical and tree sizes were similar. The number of instrumented trees decreased in 2014, and again 2015. Details of samples see **Table 1.**

**Table 1** Diameter at breast height (DBH, cm), tree height ($H$, m), height of first live branch ($H_b$, m), 1$^{st}$ quartile of DBH ($Q_1$),

3$^{rd}$ quartile of DBH ($Q_3$), sapwood width (SW, cm), sapwood area ($A_s$, cm$^2$) in 2013 to 2015. The mean values and standard deviations (S.D.) were given, the $n$ is the number of sampling trees.

| Year | DBH (cm) | | | $H$ (cm) | $H_b$ (cm) | SW (cm) | $A_s$ (cm$^2$) |
|---|---|---|---|---|---|---|---|
| | mean ± S.D. | $Q_1$ | $Q_3$ | | | | |
| 2013 ($n = 25$) | 18.0 ± 2.7 | 16.1 | 18.9 | 10.3 ± 0.7 | 4.5 ± 1.1 | 5.5 ± 0.6 | 203 ± 58.6 |
| 2014 ($n = 22$) | 17.1 ± 2.1 | 15.6 | 18.7 | 9.3 ± 0.8 | 3.7 ± 0.5 | 5.3 ± 0.5 | 182 ± 42.0 |
| 2015 ($n = 13$) | 17.7 ± 2.2 | 17.2 | 18.8 | 9.1 ± 0.8 | 3.7 ± 0.3 | 5.4 ± 0.5 | 189 ± 52.3 |

**2.3 Measurements**

**2.3.1 Micrometeorological variables**

Micrometeorological variables including solar radiation ($R_s$), net radiation ($R_n$), air temperature ($T_a$), relative humidity (RH), wind speed ($W_s$) and rainfall ($R$) were measured using an automatic weather station (AR5, Avalon Scientific, Inc. USA) lo- cated about 50 m away from the experimental field. All sensors were installed 2.0 m above the ground except the rain gauge, which was 0.5 m above the ground. Variables were measured at 1 min intervals, averaged and recorded per hour. Reference evapotranspiration ($ET_0$) was calculated using the FAO Penman-Monteith equation based on the variables $R_n$, $T_a$, RH and $W_s$

(Allen et al., 1998) at hourly base (Eq. (1)). Daily $ET_0$ was summed from hourly $ET_0$ for a day.

$$ET_0 = \frac{0.408(R_n-G)+\gamma\frac{900}{T_a+273}u_2(e_s-e_a)}{\Delta+\gamma(1+0.34u_2)} \qquad (1)$$

where $ET_0$ = reference evapotranspiration (mm h$^{-1}$),

$\Delta$ = slope of saturated water vapour pressure against air temperature $T_a$ (kPa °C$^{-1}$),

$R_n$ = net radiation (MJ m$^{-2}$),

$G$ = soil heat flux (MJ m$^{-2}$),

$\gamma$ = the psychrometric constant (kPa °C$^{-1}$),

$e_s$ = saturated vapour pressure (kPa),

$e_a$ = actual vapour pressure (kPa), and

$u_2$ = mean wind speed at 2 m height (m s$^{-1}$).

The value of vapor pressure deficit ($D$, kPa) was calculated using the following formula (Campbell and Norman, 1998):

$$D = 0.611exp^{\left(\frac{17.502T_a}{T_a+240.97}\right)}(1-\text{RH}) \qquad (2)$$

**2.3.2 Soil moisture content and groundwater table**

Volumetric soil moisture contents ($\theta$, %) were measured at depths of 0.1, 0.2, 0.4, 0.6, 0.9, 1.2, 1.6 and 2.0 m using ECH$_2$O

EC-5 sensors (Decagon Devices Inc., USA). Three placements in experiment area were measured. Each placement was set between four neighborhood sample trees. Measurements were done at 10 min intervals with hourly means recorded by a

SQ2020 data logger (Grant Instruments Ltd, UK). The sensors was calibrated using a site-specific equation based on the oven-drying method (Eq. (3)):

$\theta = 0.9677\theta_s + 0.2635 \ \ (R^2 = 0.96, n = 194, \text{RMSE} = 0.41)$         (3)

where $\theta_s$ are the output of the sensors; $\theta$ is the calibrated soil moisture content at each depth and placement. The mean $\theta$

within a certain soil layer was weight-averaged based on the depth of sensor installation. The mean field capacity ($\theta_f$) in 0-1

m soil layer of testing soil is 18.0 % by field observation. The minimum soil moisture content ($\theta_{min}$) measured during three years was 2.3 %. Relative extractable water (REW) in the upper 1.0 m soil layer was calculated using Eq. (4) (Granier,

1987).

$\text{REW} = \dfrac{\overline{\theta}_{0-1.0\ m} - \theta_{min}}{\theta_f - \theta_{min}}$     (4)

The more specific classification to quantify the degree of drought at our site is defined in **Table 2**.

Groundwater table ($g_w$) was monitored *in situ* manually once per month using a measuring tape with a cone.

**Table 2** Classification of soil drought based on relative extractable water (REW) from the measurements and preliminary reports in Mongolian Scots pine. The $T_r$ indicates transpiration rate, $C_s$ for stomatal conductance and $C_i$ for intercellular carbon oxide concentration.

| Parameter | Volumetric soil moisture content (%) | REW | Degree of drought | Description for bio-physiological variance |
|---|---|---|---|---|
| $D_0$ | $\overline{\theta}_{0-1.0\ m} > 0.4\ \theta_f$ | REW > 0.31 | No drought | Normal growth |
| $D_{mil}$ | $0.3\ \theta_f \leq \overline{\theta}_{0-1.0\ m} \leq 0.4\ \theta_f$ | $0.20 \leq$ REW $\leq 0.31$ | Mild drought | Weak growth (Jiao, 2001); $T_r$, $C_s$ and $C_i$ decreased by 46.2 %, 33.2 % and 0.9 %, respectively (Zhu et al., 2005; Tang et al., 2015); |
| $D_{mod}$ | $0.2\ \theta_f < \overline{\theta}_{0-1.0\ m} < 0.3$ | $0.08 \leq$ REW $< 0.20$ | Moderate drought | 30% leaves withered (Zhu et al., 2005); $T_r$, $C_s$ and |

| | | | | |
|---|---|---|---|---|
| $\theta_f$ | | | | $C_i$ decreased by 62.1 %, 48.6 % and 51.1 %, respectively (Zhu et al., 2005; Tang et al., 2015); |
| $D_s$ | $\overline{\theta}_{0-1.0\,m} \le 0.2\,\theta_f$ | REW < 0.08 | Severely drought | Leaves withered and some of the branch die (Jiao, 2001); $T_r$, $C_s$ and $C_i$ decreased by 70.9 %, 77.3 % and 67.6 %, respectively (Zhu et al., 2005; Tang et al., 2015) |

**2.3.3 Sap flux measurements**

Sap flux in the outermost sapwood (0–3 cm depth) ($J_{s\text{-outter}}$, cm min$^{-1}$) was measured continuously using the Granier-type thermal dissipation method (Dynamax Inc., Houston. TX. USA). Each probe was installed under the cambium on the north side of the stem at breast height (1.3 m) with pairs of probes 0.04 m apart vertically. The upper probe included a heater and the lower probe was unheated and so remained at trunk temperature for reference. Each sensor was carefully removed at the end of each growing season and reinstalled next year. The temperature difference between the upper (heated) probe and the lower (reference) probe was measured at 1-min intervals, with mean values recorded at 10-min intervals using SQ2020 data loggers. The sensors were shielded with thick aluminum-faced foam to minimize warming by radiation and exposure to rain and physical damage. The Granier empirical equation for $J_s$ was adopted as Eq. (5):

$$J_{s-\text{outter}} = 119 \times 10^{-4} \left(\frac{\Delta T_0 - \Delta T}{\Delta T}\right)^{1.231} \tag{5}$$

where $\Delta T$ is the actual temperature difference observed between heated and reference probes, and $\Delta T_0$ is the maximum $\Delta T$

value when sap flow is close to zero (generally just predawn) determined over about 10 consecutive days (Lu et al., 2004;

Dang et al., 2014).

Since the sap flux at inner part of sapwood (beyond 3 cm depth) ($J_{s-\text{inner}}$) is low due to the relative inactivity of conductive xylem, we adopted a coefficient 0.56 (Lu et al., 2004; Nadezhdina et al., 2002) for the calibration.

**2.3.4 Calculation of sap flow**

Volumetric sap flow ($cm^3 h^{-1}$) were the product of sap flux and corresponding sapwood area. At first, the sap flux measurements

$J_s$ was converted to a daily base (Eq. (6)). The daily mean $J_t$ for all measured trees in experiment area was then used to calculate daily transpiration ($T_s$, mm $day^{-1}$) of the forestry (Eq. (7)). Because all trees were at the same age and regularly-spaced, each tree was assumed to occupy equal ground per sapwood area. Hence, the ground area fraction of each tree ($A_{g,i}$, $cm^2$) was approximated as the product of individual sapwood area and the ratio of total stand sapwood area ($A_{s-stand}$, $m^2$) divided by total stand ground area ($A_{g-stand}$, $m^2$).

$$J_{t,i} = \sum_{j=1}^{24} \frac{(J_{s-outter,i} \times A_{s-outter,i} + J_{s-inner,i} \times A_{s-inner,i})A_{s,i}}{A_{g,i}} \times 60 \qquad (6)$$

$$T_s = \frac{1}{n}\sum_{i=1}^{n} J_{t,i} \qquad (7)$$

where,

$J_{s-outter,\ i}$ is the mean sap flux density in the probe-touched sapwood of a tree $i$ and $J_{s-inner,\ i}$ is probe-untouched part, $A_{s,\ i}$ is sapwood area of tree $i$ ($A_{s-outter} + A_{s-inner}$), $A_{g,\ i}$ is ground area of a tree $i$ weighted by sapwood area, $J_{t,\ i}$ is the daily sap flow of a tree $i$ (mm $day^{-1}$), $n$ is the numbers of trees measured each year.

**2.4 Statistical analyses**

The effect of soil moisture content on normalized transpiration ($T_s / ET_0$) was tested by one-way analysis of variance (ANOVA) and a Tukey HSD *post hoc* multiple comparisons test using SPSS 20 (SPSS Inc., Chicago, IL, USA). Significant correlations between $T_s$ or $T_s / ET_0$ and environmental factors over different periods were determined by Pearson's correction coefficient tests at $P < 0.05$ or 0.01. The other statistical analyses and plots employed OriginPro 2016 version 9.3 (OriginLan

Inc., Northampton, MA, USA).

**3 Results**

**3.1 Seasonal dynamics of stand transpiration and environmental factors**

The amounts of precipitation during the investigation periods were 554 mm in 2013, 384 mm in 2014 and 408 mm in 2015.

Rainfall was concentrated over quite short periods (**Fig. 4a**). The rainfall variation in experimental years in study region was high.

Daily soil moisture content exhibited large variances. The heterogeneity of soil moisture content with soil depth was significant ($P < 0.01$). In a wet year 2013, soil moisture content was higher than a dry year 2014 (**Fig. 4b**). In the second dry year 2015, there was a long drought period in July and August. After a heavy rainfall in late August (232 days), the soil of both upper and deep layers were refilled with a high soil moisture content. Based on our classifications (**Table 2**), the days of moderate and severe drought ($D_{mod} + D_s$) accounted for 19 %, 34 %, 66 % and 85 % of the whole three-year period for the four soil layers at 0–0.6, 0.6–1, 1–1.5 and 1.5–2 m, respectively. Thus, for MP it is the upper 1.0 m soil layer that provides the main water source, having the highest levels of soil moisture $\overline{\theta}_{0-0.6\,m}$ and $\overline{\theta}_{0.6-1\,m}$. Intense rainfall that infiltrated to, and thus helped recharge, the deeper layers of soil ($\overline{\theta}_{1.5-2\,m}$) were very rare from the later July in 2013 to later August in 2015.

There were no significant differences between years for daily $R_s$ ($P = 0.4$) or daily mean $D$ ($P = 0.25$) (**Figs. 4c, 4d**). The

$D$ over the whole period never exceeded 2.4 kPa (**Fig. 4d**). The groundwater table ($g_w$) at the start of the experiment was 5.2

m but significantly lowered to 5.6 m at the end of the experiment in 2015 (**Fig. 4e**).

The daily $T_s$ showed similar seasonal patterns with $R_s$ (**Fig. 4f**), indicating the radiation was a major factor to affect plant transpiration. Overall, $T_s$ was at a relative high level in May each year until August, and then gradually decreased to a low level in late October. The seasonal dynamics of $T_s$ reflected the variations in physiological traits of MP and meteorological factors. The $T_s$ between the years was significantly different ($P < 0.01$). The maximum daily $T_s$ in 2013 was 3.73 mm day$^{-1}$

with a mean value of 1.94 mm day$^{-1}$ during growing season. In 2014, the maximum daily and seasonal average $T_s$ were de- creased to 2.55 mm day$^{-1}$ and 1.46 mm day$^{-1}$. In 2015, the great decrease of maximum $T_s$ occurred down to a value of 1.40 mm day$^{-1}$, as well as for the seasonal mean down to 0.81 mm day$^{-1}$ (**Fig. 4f**). The decreasing $T_s$ between seasons was partially due to less rainfall and water availability, and probably also due to the plant recovery capability in relation to the permanent changes in plant physiological traits.

[Figure]

**Figure 4** Seasonal time courses of precipitation, mean volumetric soil moisture content ($\overline{\theta}$), solar radiation ($R_s$), vapour pressure deficit ($D$), groundwater table ($g_w$), and daily mean sap flux of stands ($T_s$) in 2012 to 2015. The grey area in (b) indicates moderate and severe drought.

**3.2 Normalized transpiration affected by soil relative extractable water**

The normalized transpiration $T_s / \mathrm{ET_0}$ under sufficient water supply $D_0$ was about $0.29 \pm 0.09$ (**Fig. 5a**), 15% higher than under mild drought, 62% higher than under moderate drought and 149 % higher than under severe drought. The maximum ratio of daily sap flux to reference evapotranspiration ($T_s$ / $ET_0$) increased with REW sharply at low REW but keep constant when the

REW was above 0.31 (**Fig. 5b**). This indicated that there are the other factors besides the atmospheric and soil moisture affected the water use of MP, in which the stomatal regulation or the seasonal variation of biological rhythms is important one.

[Figure]

**Figure 5** Normalized transpiration $T_s$ / $ET_0$ affected by soil droughts. Normalized sap flux by using reference evapotranspiration indicates a potential transpiration ability under maximum evaporative demand caused by metrological factors, the relationship between $T_s$ /$ET_0$ and relative extractable water (REW) is mainly affected by plant traits. The maximum of $T_s$ / $ET_0$ at the REW step of 0.02(dimensionless) are selected out (red circles in (b)) and modelled by an exponential function. The dashed line is at REW=0.31. Values of $T_s$ / $ET_0$ followed by different letters are significantly different at $P < 0.05$ by univariate ANOVA (post hoc Tukey HSD).

**3.3 Progressive decline of sap flux with developing of drought and recovery following rain**

The 49-day periods from DOY 203 to 251 each year was chosen to illustrate the changes in $T_s$ with REW, a dry-wet shift. In the period of wet year (2013), the soil moisture is always in $D_0$ (without water stress) and mean $T_s$ was about 2.15 mm day$^{-1}$

(**Fig. 6a**). In the first dry year (2014), $T_s$ decreased by as much as 18 % under a mild water stress ($D_{mil}$) and further by 40 %

under moderate stress ($D_{mod}$). The $T_s$ was greatly recovered after a heavy rain (**Fig. 6b**). In the second dry year (2015), the $T_s$

decreased by 73 % under $D_{mod}$ and further by 74 % under $D_s$ stage (**Fig. 6c**). The daily $T_s$ under $D_s$ was only 0.55 mm day$^{-1}$.

This very little transpiration likely only sufficient to maintain the survival of MP. After a heavy rainfall, even the soil water status was improved a lot, the $T_s$ of trees was still very low (less than 1.4 mm day$^{-1}$), indicating the $T_s$ of MP was difficult to be recovered.

[Figure]

**Figure 6** The comparison of measured transpiration ($T_s$) and relative extractable water (REW) in the upper soil layer (above 1 m) during maximum growth period from DOY 203 to 251 in 2013 to 2015. $D_0$, $D_{mil}$, $D_{mod}$, and $D_s$ are no, mild, moderate and severe droughts, respec- tively (see Table 2). The increase of REW was due to the rainfall.

**4 Discussion**

**4.1 Reduction of soil moisture content under droughts**

In our site, the long term trends for increasing air temperature and decreasing annual precipitation (**Fig. 2**) is unfavorable to the growth of trees. The declining groundwater and the coarse sandy soil (>83 % sand particles in our site) prevented capillary ascension efficiently (less than 0.5 m) (Vincke and Thiry, 2008). Sandy soils have low water holding capacity and high hy- draulic conductivity, thus water percolates through this soil quickly after a rain. During the three-year periods in our site, there are an effective rain event every 14 days averagely (rainfall intensity is more than 10 mm per times). Under well-wetted soil conditions ($D_0$), $\overline{\theta}_{0-1.0\,m}$ was depleted at the high rate of 1.9 vol % per day during the first two days and at the rate of 0.35 vol %

per day during the subsequent nine days (**Fig. 7**) because of either soil water holding capacity or great water uptake by trees.

The depleting rate of $\overline{\theta}_{0-1.0\text{ m}}$ under the drought conditions was only 0.09 vol % per day under severe drought. That indicates the only little of water was absorbed by trees under severe water stress. Our results suggested the plant might adjust their physiological traits, e.g. closing stomatal and reduing root system to at first priority for the survival. The sap flux declines very quickly in desiccated root system (Mereu et al., 2009).

[Figure]

**Figure 7** There will be an effective rain event every 14 days averagely in our site (Rainfall intensity is more than 10 mm per times), which acted as a window to analyze the decrease rate of soil water during this period. Scatter-line plot described the relationships between decrease rate of upper soil moisture ($\overline{\theta}_{0-1.0\text{ m}}$) and time under different initial degree of drought levels which were defined in **Table 2**.

**4.2 Contribution of water in the upper and deep soil layers**

The MP is a shallow-rooted species with root density decreasing sharply below 1.0 m (Jiang et al., 2002; Zhu et al., 2005; Zhu et al., 2008), implying the soil moisture in the upper 1.0 m layer provides major water source for transpiration (Su et al., 2006;

Wei et al., 2013; Song et al., 2014). In our study, the rapid recovery of $T_s$ when $\overline{\theta}_{0-1.0\text{ m}}$ was increased after a rain (**Figs. 4 and**

**6**) suggested that MP was very sensitive to the changes in the available water in the upper soil layer. Uptake by the shallow roots decreased very significant as this soil layer dried out (**Fig. 6**). However, under severe drought, for example in August of

2015, the MP trees used quite amount of deep soil water. It might be carried out by developing more letaral root system in deep soil. The fine roots of Scots pine die quickly under drought conditions (Vanguelova and Kennedy, 2007). Therefore, it would cause a death of new developed fine root system, resulting a permanent declining in the capability of transpiring water even when the soil water status was improved (**Fig. 4**). The death of fine root in deep soil layer may explain why after a rainfall, post-stress sap flux recovery is very small after a long and severe drought in 2015.

**4.3 Groundwater is an important source for plant adaptation under long and severe drought**

Mongolian Scots pine is a dimorphic-rooted species where, the maximum taproot depth in a sandy soil can up to 2.7 m (Cana- dell et al., 1996), and even to 5.2 m for a 42-year-old tree in a sandy soil near our site (Jiang et al., 2002). Our results on the depletion of soil water in 1.5-2 m soil layers, existed but not large, also suggested a deep taproot depth in MP. This enables

MP trees to use deeper water source (i.e. groundwater), especially under drought (Barbeta et al., 2015; Hentschel et al., 2016).

This is likely to occur when soil moisture content in the upper soil layers (0–60 cm) declines to 3.6 % (Wei et al., 2013; Song et al., 2016a). In our site, $g_w$ lowered from $5.03 \pm 0.14$ m in 2013 to $5.47 \pm 0.09$ m in 2015 (**Fig. 4e**). From late 2014, the value of $g_w$ was always far deeper than 5.2 m and thus unlikely accessible directly by our instrumented trees if their tap roots were shallower than 5.2 m. However, in the severe drought ($D_s$, with minimum $\overline{\theta}_{0-1.0\,m}$ as low as 2.3 %), we recorded a clear diurnal pattern of sap flux with the much reduced daily $T_s$ (mean 0.56 mm day$^{-1}$, or 28.2 % of that for $D_0$). Hence, we inferred that significant groundwater contributions to $T_s$ occurred only under severe drought conditions though determining just what pro- portion of that water came from the groundwater or from tree storage is beyond the scope of this study. It has been reported that as rainfall decreases, tree dependence on groundwater increases (Kume et al., 2007).

[Figure]

**Figure 8** Normalized transpiration ($T_s$ / ET$_0$) in Mongolian Scots pine affected by the groundwater table ($g_w$) .

**4.4 Transpiration of the plantation and implications**

There is a complex interplay between the various meteorological factors, e.g. solar radiation, vapour pressure deficit, air temperature, wind speed and relative humidity, and directly or indirectly influences transpiration in a tree. These variables were aggregated into a variable $ET_0$, which serves as an index of atmospheric water demand power (Zha et al., 2010). Therefore, as expected, changes in $ET_0$ trigger a prompt plant response in terms of transpiration. Changes in precipitation (and hence soil mositure) affect transpiration but likely over a long temporal scale (Yan et al., 2016). Our results also showed a strong reduction in normalized transpiration $T_s / ET_0$ mostly after a long period drought. Using normalized transpiration allows to focus on the effects of soil water availability and plant physiological responses. This behavior has also been found in Scots pine in Europe (Poyatos et al., 2005), presenting the strong effects of stomatal regulation for controlling the rate of water loss. The significant fall in $g_w$ seems to explain the difficulty in plant recovery of $T_s$ after a heavy rain.

The reduction in transpiration of MP due to soil water shortage was 25% in first dry year and 58% in the second dry year (**Table 3**). This is comparable with reported values of 40 to 80 % for different species in different habitats (Leuzinger et al., 2005; Gartner et al., 2009; Betsch et al., 2011). Average cumulative $T_s$ values in testing MP during a whole growing season ranged from 145 to 357 mm. This was higher than in a sparse forest of 150-year-old Scots pine growing in an inner Alpine dry valley (Wieser et al., 2014) and in northeastern Germany (Lüttschwager et al., 1998) due to the larger canopy size and environmental conditions. In this study, the annual water transpiration by MP in wet year (357 mm) is nearly equal to 75 % of the total annual precipitation in a historical normal year. However, considering the soil evaporation, transpiration by understory plants (e.g. weeds), and leaf interception and vaporization, the current stand density of 625 trees per ha is likely too high for a sustainable ecosystem of Mongolian Scots pine forestry.

Transpiration in a coniferous forest is often conservative with relatively low values of canopy conductance (Levitt, 1980). For instance, Scots pine has a rather conservative water use strategy with a very plastic response to intermittent dry periods with high use of stored water (Arneth et al., 2006; Verbeeck et al., 2007). In our study, we found MP was more moderate in its water consumption than many broad-leaved forest tree species growing nearby (e.g. *Populus* spp) (Zhu et al., 2005). Although the groundwater table decreases in our experiment, the MP still contributes less to the groundwater table decline than the more extensive and/or intensive agricultural land uses (0.1 m per year) (Song et al., 2016b). The lateral roots of an MP tree can extend laterally to about 0.65-times tree height (Jiang et al., 2002; Su et al., 2006). This helps MP to obtain water from the upper soil layers efficiently (Song et al., 2014). The ability of MP to maintain a low sap flux even during severe drought suggests a strong adaptation under climate change (Waromg et al., 1979), especially when the extreme weather events increase in the future. However, the advanced mature period was found when Mongolian Scots pine introduced from the north (origin distribution region) to south (planted region, this study area) (Jiao et al. 1989; 2001). The difficulty in recovery for water uptake by 30 years MP trees under severe drought might also caused by the low growth vigor of old trees. It implies that the re-forestry might be necessary when MP trees are over 30 years old.

**5   Conclusions**

Mongolian Scots pine was relatively conservative in water use with a maximum of 3.73 mm day$^{-1}$. Stand transpiration during the growing season ranged from 149 mm in an extreme dry year to 357 mm in a wet year. The sap flux in MP responded strongly to soil water availability. The daily sap flux reduced with drought by 74% as the duration and intensity of drought was high in dry years. The ability of recovery in plant transpiration was limited by the duration and severity of drought. Our results suggest that the degradation in MP plantation is attributable to the combined effects of large temporal variation in rainfall and the ability of specific recovery after the occurrence of drought. The results could help farmer improve the management and sustainability of MP forestry by optimizing plant density and reforestation in semi-arid region.

**Acknowledgements**

[revised manuscript text omitted]

Nadezhdina, N., Cermak, J. and Ceulemans, R.: Radial patterns of sap flow in woody stems of dominant and understory species:

scaling errors associated with positioning of sensors, Tree Physiol., 22, 907-918, 2002.

Poyatos, R., Llorens, P. and Gallart, F.: Transpiration of montane *Pinus sylvestris* L. and *Quercus pubescens* Willd. forest stands measured with sap flow sensors in NE Spain., Hydrol. Earth Syst. Sc., 9, 493–505, 2005.

Song, L.N., Zhu, J.J., Li, M.C. and Yu, Z.Y.: Water utilization of *Pinus sylvestris* var. *mongolica* in a sparse wood grassland in the semiarid sandy region of Northeast China, Trees, 28, 971–982, 2014.

Song, L.N., Zhu, J.J., Li, M.C., Zhang, J.X. and Lv, L.Y.: Sources of water used by *Pinus sylvestris* var. *mongolica* trees based on stable isotope measurements in a semiarid sandy region of Northeast China, Agr. Water Manage., 164, 281–290, 2016a.

Song, L.N., Zhu, J.J., Li, M.C. and Zhang, J.X.: Water use patterns of *Pinus sylvestris* var. *mongolica* trees of different ages in a semiarid sandy lands of Northeast China, Environ. Exp. Bot., 129, 94–107, 2016b.

Su, F.L., Liu, M.G., Guo, C.J. and Zhang, Q.: Characteristics of vertical distribution of root system of Mongolian scots pine growing in sandy area and influence to the soil, Soil and Water Conservation in China, 286, 20–23, 2006 (in Chinese with

English abstract).

Su, H., Li, Y.G., Liu, W., Xu, H. and Sun, O.J.: Changes in water use with growth in *Ulmus pumila* in semiarid sandy land of northern China, Trees, 28, 41–52, 2014.

Tang, F.D., Lin, Y. and Li, Y.: Impact of water stress photosynthesis characteristics of Mongolian pine seedlings and Grafting

Korean pine seedlings with stocks of Mongolian pine, Journal of Liaoning University, 42, 274-276, 2015 (in Chinese with English abstract).

Vanguelova, E.I. and Kennedy, F.: Morphology, biomass and nutrient status of fine roots of Scots pine ( *Pinus sylvestris* ) as influenced by seasonal fluctuations in soil moisture and soil solution chemistry, Plant Soil, 270, 233–247, 2007.

Verbeeck, H., Steppe, K., Nadezhdina, N., De Beeck, M.O., Deckmyn, G., Meiresonne, L., Lemeur, R., Cermak, J., Ceulemans,

R. and Janssens, I.A.: Model analysis of the effects of atmospheric drivers on storage water use in Scots pine, Biogeosci- ences, 4, 657–671, 2007.

Vincke, C. and Thiry, Y.: Water table is a relevant source for water uptake by a Scots pine (*Pinus sylvestris* L.) stand: Evidences from continuous evapotranspiration and water table monitoring, Agr. Forest Meteorol., 148, 1419–1432, 2008.

Waromg, R.H., Whitehead, D. and Jarvis, P.G.: The contribution of stored water to transpiration in Scots pine, Plant, Cell &

Environment, 2, 309–317, 1979.

Wei, Y.F., Fang, J., Liu, S., Zhao, X.Y. and Li, S.G.: Stable isotopic observation of water use sources of *Pinus sylvestris* var.

*mongolica* in Horqin Sandy Land, China, Trees, 27, 1249–1260, 2013.

Wesche, K., Walther, D., Wehrden, H.V. and Hensen, I.: Trees in the desert: Reproduction and genetic structure of fragmented

*Ulmus pumila* forests in Mongolian drylands, Flora, 206, 91–99, 2011.

Wieser, G., Leo, M. and Oberhuber, W.: Transpiration and canopy conductance in an inner alpine Scots pine (*Pinus sylvestris*

L.) forest, Flora - Morphology, Distribution, Functional Ecology of Plants, 209, 491–498, 2014.

Yan, M.J., Zhang, J.G., He, Q.Y., Shi, W.Y., Otsuki, K., Yamanaka, N. and Du, S.: Sapflow-based stand transpiration in a semiarid natural oak forest on China's Loess Plateau, Forests, 7, 227–240, 2016.

Yuan, M.W., Zhang, L.Z., Gou, F., Su, Z., Spiertz, J.H.J., van der Werf, W.: Assessment of crop water productivity in semi- arid Inner Mongolia. Agr. Water Manage., 122, 28–38, 2013.

Zha, T.S., Barr, A.G., Kamp, G.V.D., Black, T.A., Mccaughey, J.H. and Flanagan, L.B.: Interannual variation of evapotran- spiration from forest and grassland ecosystems in western Canada in relation to drought, Agr. Forest Meteorol., 150,

1476–1484, 2010.

Zheng, X., Zhu, J.J., Yan, Q.L. and Song, L.N.: Effects of land use changes on the groundwater table and the decline of *Pinus*

*sylvestris* var. *mongolica* plantations in southern Horqin Sandy Land, Northeast China, Agr. Water Manage., 109, 94–

106, 2012.

Zhu, J. J., Li, F.Q., Xu, M.L., Kang, H.Z. and Wu, X.Y.: The role of ectomycorrhizal fungi in alleviating pine decline in semiarid sandy soil of northern China: an experimental approach, Ann. Forest Sci., 65, 304, 2008.

Zhu, J.J., Kang, H.Z., Li, Z.H., Wang, G.C. and Zhang, R.S.: Impact of water stress on survial and photosynthesis of Mongolian pine seedlings on sandy land, Acta Ecologica Sinica, 25, 2527-2533, 2005 (in Chinese with English abstract).

Zhu, J.J., Zeng, D.H., Kang, H.Z., Wu, X.Y. and Fan, Z.P.: Decline of *Pinus sylvestris* var. *mongolica* plantations on Sandy

Land, Chinese Forestry Press, Beijing, 264 pp, 2005 (in Chinese with English abstract).

---

## Author Comment (AC3) · 19 Apr 2017

Genera comments: The MS by the authors presents data from 3 years of measurements of Mongolian Scots Pine. While such data will be very useful for our understanding of this species plant water use and strategies, as well as ecosystem functioning in which this species is found, the collection of sap flow data and subsequent statistical analyses were poor and do not warrant publication.

Response: We greatly appreciate the critical and very helpful comments raised by the referee. We carefully discussed the comments with co-authors and thoroughly revised the manuscript as in supplement.

Specific comments: The authors deployed the Granier technique, or thermal dissipation probe (TDP), to measure sap flow in Mongolian Scots Pine. The TDP sensor only has one measurement point that the authors installed at an unknown depth in the sapwood. The width, or depth, of the sapwood is also unknown. From the single measurement point, sap flow data were scaled upwards to tree and then entire forest stand. Such a scaling approach no doubt introduced significant errors as it is known, from a vast amount of research, that sap flow varies across the radial profile of sapwood. Measuring at a single point in the sapwood will then significantly over- or under-estimate true sap flow. Furthermore, the Granier technique is an empirical technique and requires a species specific calibration. The authors chose a generic calibration which may lead to further errors in their estimate of sap flow. Therefore, the sap flow data collected by the authors cannot reliably be used to estimate total tree or stand water use.

Response: In this study, we measured sap flux density ($J_s$) by Granier-type thermal dissipation method (Dynamax Inc., Houston. TX. USA) with 3-cm probe. The sapwood width of the trees in our experiments ranged from 4.38 to 6.98 cm for all years and trees, We added this information in the revised text. The probes were thus installed only in sapwood properly.

We totally agree with the referee on the necessity of the species-specific calibration of TDP where possible. In this paper, we used a generic calibration considering the sensors were all in active xylem (sapwood). Granier and other early users demonstrated that the generic calibration produced total sap flow within an error of 10% and smaller based on many different type of xylem structure including Pinus (Goulden ML, Field CB. 1994; Lu et al, 2004; Do, et al,1998). The validity of the original calibration has also been supported by results from studies showing generally good quantitative agreement between water flux estimates obtained with the Granier method compared with other sap flow methods, water absorption, branch bag measurements, eddy covariance measurements and catchment-scale water balance (Granier et al. 1990, 1994;

Kostner et al. 1996; Saugier et al. 1997; Tournebize and Boistard, 1998; Ewers et al. 2007; Ford et al. 2007; Lu etal, 2004). Considering most of validation for TDP is made on excised stems under very strict laboratory controlling conditions, we use the original Granier generic parameter before a proper one developed in future work.

The calculation of total transpiration by TDP in 0-3 cm radial is limited by how much this data represent the whole tree. Previous studies in similar tree species (Pinus sylvestris) showed that the sap flux within 0-3 cm sapwood width covers 64% of total sap flux in radial direct for a tree (Lu et al., 2004; Nadezhdina et al., 2002). To calibrate the transpiration of TDP measurements, we separated sapwood into outer area (0-3 cm) and inner area (3 cm to the heartwood), a coefficient as the ratio of the sap flux in inner area by that in outer area was used with a value of 0.56 (Lu et al., 2004; Nadezhdina et al., 2002)(Fig.1). Accordingly, the data of sap flux measurements in our study were calibrated by this coefficient in the revised text. We also clarified in the M&M section.

Comment: The statistical approach by the authors is also incorrect. The authors analysed a series of collinear variables across a series of univariate or a multivariate regression analysis. All the environmental variables are related to each other therefore the author's approach will introduce a significant amount of error. Collinearity in regression models is a commonly overlooked statistical error. I recommend the author's read the chapter on collinearity in Quinn and Keough (2002) "Experimental Design and Data Analysis for Biologists". The authors can run their predictor variables through a Principle Components Analysis to reduce their related variables to a series of unrelated variables. The factor scores that come out of the PCA can then be used in a multiple linear regression analysis as the predictor variables.

Response: We totally agree. In the revised text, we use a normalized transpiration, $T_s/ET_0$, the potential transpiration under actual evaporative demand determined by meteorological factors, e.g. solar radiation, air temperature, humidity and wind speed (Yuan et al., 2013). Thus, the relationship between normalized transpiration and soil relative extractable water (REW) is not in collinearity. The parameters of this relationship reflect the plant response (changing in physiological traits) to the water availability. We deleted the multivariate regression between, gw, REW and D.

Comment: The analyses and results have the feeling that the authors are fishing through their data set looking for significant patterns and then presenting those patterns. There is no systematic analytical approach and it is extremely difficult to follow and comprehend. For example, several drought periods are declared based on REW and a regression analysis between Ts and ET0 is presented in Figure 4. But what is the significance of REW <= 0.24? Why is it not 0.25 or 0.23? These REW values may be important but it just seems some random numbers were chosen. The MS would be far stronger, and much easier to read and comprehend, if there was a systematic and biologically realistic designation of drought periods.

Response: We provide the revised data about REW levels in Table 2 and listed out the reference. We focused on the biological traits such as growth status and physiological index such as transpiration rate of needles (Tr), stomatal conductance (Cs) and intercellular carbon oxide concentration (Ci) to divide drought periods and give a designation.

Comment: Figure 5 also suffers from this problem. What is EW? Is this something the authors have just created? If it is so important, why is it being introduced deep into the Results section and not in the Introduction section? Why is the power model important? Does it have a slightly better R2 than a linear regression model? But does it carry greater explanatory power in terms of an AIC analysis or other statistical approach?

Response: We agreed. In the revised text, we deleted this.

Comment: Throughout the MS, there are several grammatical and spelling errors. For example, "sapflow" should be "sap flow".

Response: We thoroughly revised the manuscript and corrected those mistakes.

Other points: The linear regression in figure 1 is in appropriate and statistically incorrect. Is the purpose of the regression to demonstrate that rainfall decreased and temperature increased over this time period? If yes, probably consult the climate science literature to determine the methods they use to statistically quantify such changes.

Response: We agreed. We use the anomalies of air temperature and precipitation (%) to display the fluctuation of variables and trends during the long period.

Comment: Can you please provide more details about the soil profile – for example, is the texture consistent down the profile or are there distinct horizons? Is the texture described in the text only for the top 40cm of soil where, presumably, most of the MP root activity is?

Response: We added related information about soil profile in Line 82-85 in revised manuscript, that is: The soil is sandy with a sedimentary aeolian sand layer more than 3 m and an ancient alluvial sand layer with the total depth more than 126 m (Jiao, 1989). The mean bulk density of the upper 2 m soil layer is 1.61 g cm-3. The mean soil texture is 83 % of sand (> 0.05 mm), 9 % of silt (0.05–0.002 mm) and 8 % of clay (< 0.002 mm). The organic matter content is 0.3–1.0 g kg-1.

Comment: Where were the soil moisture sensors installed? Next to the weather station outside of the plot? or somewhere inside of the plot?

Response: Three placements in experiment area were measured. Each placement was set between four neighborhood sample trees. Measurements were done at 10 min intervals with hourly means recorded by a SQ2020 data logger (Grant Instruments Ltd, UK). We clarified this in the revised text.

Comment: What was the sapwood width of MP?

Response: We added the data of sapwood width (SW, cm) of MPs in Table 1.

Comment: Why were the TDP sensors installed on the north-face of the trees? Presumably, this is where sap flow would be lowest around the circumference of the trunk.

Response: The reason of the TDP sensors installed on the north-face of the trees is mainly to minimize interference from solar direct radiation in northern hemisphere.

We added four references, see below, to the manuscript in Line 381-382, 396-397, 416-417, 426-427and also into references part.

Nadezhdina, N., Cermak, J. and Ceulemans, R.: Radial patterns of sap flow in woody stems of dominant and understory species: scaling errors associated with positioning of sensors, Tree Physiol., 22, 907-918, 2002. Tang, F.D., Lin, Y. and Li, Y.: Impact of water stress photosynthesis characteristics of Mongolian pine seedlings and Grafting Korean pine seedlings with stocks of Mongolian pine, Journal of Liaoning University, 42, 274-276, 2015 (in Chinese with English abstract). Yuan, M.W., Zhang, L.Z., Gou, F., Su, Z., Spiertz, J.H.J., van der Werf, W.: Assessment of crop water productivity in semi-arid Inner Mongolia. Agr. Water Manage., 122, 28-38, 2013. Zhu, J.J., Kang, H.Z., Li, Z.H., Wang, G.C. and Zhang, R.S.: Impact of water stress on survial and photosynthesis of Mongolian pine seedlings on sandy land, Acta Ecologica Sinica, 25, 2527-2533, 2005 (in Chinese with English abstract).

Please also note the supplement to this comment:
http://www.biogeosciences-discuss.net/bg-2017-69/bg-2017-69-AC3-supplement.pdf

―――――――――――――――――――

[Figure]

*Radius of stem of Pinus sylvestris (relative)*

**Fig. 1.** Radial pattern of sap flux density of Pinus sylvestris. (A): from Lu et al.(2004), (B) :from Nadezhdina et al., (2002).

---

## Referee Comment (RC5) · Anonymous Referee #3 · 2 May 2017

I find that six authors have put considerable effort into addressing the comments of the referees. As a result the paper is much improved and I have no problem in recommending it for publication.

---

## Referee Comment (RC6) · Anonymous Referee #2 · 14 May 2017

I thank the authors for their thorough revision of the MS and responses to my earlier concerns. However, the most significant concern remains. The MS, in its current format, cannot address this concern therefore the paper still requires significant revision or it should be rejected.

The authors scale sap flow from tree to stand measurements and the majority of the results are then presented as Ts (stand sap flux, mm). However, there is a significant level of uncertainty in the measurement of sap flow and, consequently, there can be little confidence that the stand values are accurate.

The presentation of the sap flow data as Ts (mm) should be omitted from publication for the following reasons:

The authors deployed a TDP style sap flow sensor which only has a single measurement point at some point along the radial profile. The sensor is a 30mm length sensor but this does not mean sap flow is measured over the entire length of 30mm, but at some point along the length. Therefore, the first uncertainty is the actual zone, or area, of sapwood the sensor is sampling within that 30mm length. The sapwood needs to be measured at multiple points along the radial profile for accurate scaling of data from point measurements to tree and then stand measurements. Measuring one point along a ∼6cm radii is inadequate.

The second point of uncertainty is the TDP sensor is uncalibrated and the authors did not attempt a calibration of their sensors. By the authors' own admission, there is at least a 10% error based on a "standard" calibration equation. The authors cite a series of publications to support the 10% error value but I can easily cite other publications where error values have been reported as high as 60% (e.g. Steppe et al 2010). And that is the point. Without a calibration, we do not know whether it is a 10% error, 60% error or something else. If the authors want to present sap flow as an absolute number, in mm, then they should have calibrated the sensors.

A third point of uncertainty is the azimuthal variability of sap flow in the sampled trees. With a single measurement point, installed on one side of the tree only, it is highly uncertain whether this captures the actual sap flow of the entire sapwood cross-section of the tree. The authors should have attempted to measure and describe the azimuthal variation in sap flow, or lack thereof, to relieve this uncertainty.

Therefore, the presentation of Ts in the current MS is, at best, a guess and the values are highly uncertain.

However, the authors can still publish their data as relative sap flow rather than absolute sap flow. Rather than presenting the sap flow results as an absolute value in mm, it is possible to present the results as a relative value or percentage value. The highest value of sap flow, across the measurement period, is the 100% maximum rate and

all the other sap flow measurements are relative to this value. Presenting the sap flow data as relative sap flow rate will overcome the issue of uncalibrated sensors and uncertainty in data values.

Any reference or presentation of data throughout the MS as Ts (mm) should be removed.

Any calculation of stand scale transpiration should be removed (e.g. section 4.5).
* * *

---

## Author Comment (AC4) · 17 May 2017

Comments:

I thank the authors for their thorough revision of the MS and responses to my earlier concerns. However, the most significant concern remains. The MS, in its current format, cannot address this concern therefore the paper still requires significant revision or it should be rejected. The authors scale sap flow from tree to stand measurements and the majority of the results are then presented as Ts (stand sap flux, mm). However, there is a significant level of uncertainty in the measurement of sap flow and, consequently, there can be little confidence that the stand values are accurate. The presentation of the sap flow data as Ts (mm) should be omitted from publication for the following reasons: The authors deployed a TDP style sap flow sensor which only has a single measurement point at some point along the radial profile. The sensor is a 30mm length sensor but this does not mean sap flow is measured over the entire length of 30mm, but at some point along the length. Therefore, the first uncertainty is the actual zone, or area, of sapwood the sensor is sampling within that 30mm length. The sapwood needs to be measured at multiple points along the radial profile for accurate scaling of data from point measurements to tree and then stand measurements. Measuring one point along a âĹij6cm radii is inadequate. The second point of uncertainty is the TDP sensor is uncalibrated and the authors did not attempt a calibration of their sensors. By the authors' own admission, there is at least a 10% error based on a "standard" calibration equation. The authors cite a series of publications to support the 10% error value but I can easily cite other publications where error values have been reported as high as 60% (e.g. Steppe et al 2010). And that is the point. Without a calibration, we do not know whether it is a 10% error, 60% error or something else. If the authors want to present sap flow as an absolute number, in mm, then they should have calibrated the sensors. A third point of uncertainty is the azimuthal variability of sap flow in the sampled trees. With a single measurement point, installed on one side of the tree only, it is highly uncertain whether this captures the actual sap flow of the entire sapwood cross-section of the tree. The authors should have attempted to measure and describe the azimuthal variation in sap flow, or lack thereof, to relieve this uncertainty. Therefore, the presentation of Ts in the current MS is, at best, a guess and the values are highly uncertain. However, the authors can still publish their data as relative sap flow rather than absolute sap flow. Rather than presenting the sap flow results as an absolute value in mm, it is possible to present the results as a relative value or percentage value. The highest value of sap flow, across the measurement period, is the 100% maximum rate and all the other sap flow measurements are relative to this value. Presenting the sap flow data as relative sap flow rate will overcome the issue of uncalibrated sensors and uncertainty in data values. Any reference or presentation of data throughout the MS as Ts (mm) should be removed. Any calculation of stand scale transpiration should be removed (e.g. section 4.5).

Response:

We greatly appreciate the helpful comments raised by the referee. We carefully discussed the comments with co-authors and thoroughly revised the manuscript. We totally agreed with the referee on the potential error sources when estimating the sap flux with TDP, e.g from radial variation, azimuthal variation and calibration. In our manuscript, we attempt to consider the radial variation through measuring Js,outter and estimating Js,inner with a coefficient from Scots pine while it might be not accurate enough than measurement with multiple points sensor(e.g. HFD method from ICT ). Our experiment forest is regular planted with the canopy openness about 0.24, which helps to form uniform trees and assumed be less variation along azimuthal direction. Since the estimation of total transpiration of plantation are very necessary for exploring causes of the degradation and providing suggestion for stand density adjustment in our sandy regions, we adopted TDP, a simple and affordable device for large samples, to estimate the transpiration of a forest although some error or uncertainty remains. In spite of this, we review our data analysis again carefully and we thank the referee greatly for your alternative ways to overcome the issue of uncalibrated sensors and uncertainty in data value by presenting the results of sap flux as a relative value. In the revised text, we normalized Ts (sap flux dividing by the maximum value over three-year experiment period) to replace an absolute value to avoid the confusion. The revised manuscript provided in supplement.

Please also note the supplement to this comment:
http://www.biogeosciences-discuss.net/bg-2017-69/bg-2017-69-AC4-supplement.pdf

[Figure]

**Supplement:**

**Severe drought greatly reduces sap flux of Mongolian Scots pine (*Pinus sylvestris* var. *mongolica*) and its recovery ability in a sandy and semi-arid environment**

HongZhong DANG[1,a], LiZhen ZHANG[2], WenBin YANG[1], JinChao FENG[1], Hui HAN[3], Wei LI[1]

[1] Institute of Desertification Studies, Chinese Academy of Forestry, Beijing, 100091, China

[2] Institute of Resources and Environment, China Agricultural University, Beijing, 100094, China

[3] Institute of Sand Fixation and Afforestation of Liaoning Province, Fuxin, 123000, China

[a] *Correspondence to*: HongZhong DANG (hzdang@caf.ac.cn)

**Abstract.** Trees growing in water limited ecosystems are often exposed to the significant challenges of soil water stress due to low precipitation and high variation. In this study, we aimed to quantify the sap flux of Mongolian Scots pine (*Pinus sylvestris* var. *mongolica*) growing on a sandy soil, in a region characterised by an erratic rainfall pattern. Measurements were made over three successive years of contrasting annual rainfall - a wet year (2013), a dry year (2014), and a second dry year (2015). Over the three years, sap flux density ($J_s$) was measured at outter 3 cm width conductive xylem of 25 tree samples, then were up-scaled to daily transpiration at tree- and plot-level ($T_s$). Due to the high variation of rainfall in three years, the measurements reflected the tree response to wide range of water stress from wet (2013), mild-drought (2014) to severe-drought (2015). Generally, the normalized $T_s$ during growing seasons decreased by 25% in dry year 2014 and 58% in second dry year 2015. Stand transpiration fluctuated widely over quite short time scales (months or weeks) due to the erratic rainfall and sandy soil coupling with a declining groundwater table. Particularly, under a long-period of drought stress in late season in 2014 and early season in 2015, transpiration of Mongolian Scots pine have been restricted greatly, and the recovery of $T_s$ following heavy rainfall was incomplete (63–69%). Our results help elucidate the interplay between the effects of the atmosphere and soil moisture on tree sap flux, and highlight the negative effects of drought on sap flux of mature forest tree. Our findings provide the evidence for the observed premature degradation of these Mongolian Scots pine plantations in terms of an eco- hydrological perspective.

**Keywords:** sap flux; Mongolian Scots pine (*Pinus sylvestris* var*. mongolica* Litv*)*; soil water availability; water stress; sandy soils; semi-arid climate.

**1 Introduction**

Reforestation has been used widely in semi-arid areas to control soil erosion, to capture carbon and to serve as wind breaks (D'Odorico and Porporato, 2006). However, trees growing in severely water-limited ecosystems are often exposed to significant challenge due to insufficient soil water (Wesche et al., 2011; Su et al., 2014). Many factors influence the amount of soil water and its availability to vegetation, for instance, the amounts and timings of rainfall interval, soil water capacity, root water-uptake capacity and the availability of alternative water sources such as groundwater (Meinzer et al., 2006). Under climate change, the increasing in the frequency and severity of drought decreases soil water availability in the future (Leo et al., 2013). This would increase tree mortality rate through excessive competition for water and thus influences the structure and function of forest ecosystems (Barbeta et al., 2015). Quantification of sap flux by trees at individual and forest levels could help us to understand how environmental factors affect their water usage. It is necessary to properly assess the impacts of climate change on ecological and hydrological processes in the fragile ecosystems (Bovard et al., 2005). This knowledge would allow us to make better forest establishment decisions and management actions.

Mongolian Scots pine (*Pinus sylvestris* var. *mongolica*, MP), a geographical variety of Scots pine (*P. sylvestris*), is naturally widely distributed in northern China and in parts of Russia and Mongolia. It is found in the Daxinganling Mountains (50 °10 ′–53°33′ N, 121°11′–127°10′ E) and in Honghuaerji on the Hulun Buir sandy plains of the northeast (Zhu et al., 2008; Zheng et al., 2012). The MP is a popular species for reforestation in northern China due to its traits of good drought and cold resistance. Consequently, more than $6.7 \times 10^5$ ha of MP plantations have been established to control desertification, in the great project of the Three-North Shelter Forest Program (TNSFP) launched in China from 1978 (Zheng et al., 2012). Unfortunately, serious degradation and considerable concern has occurred in these plantations since the mid-1990s, such as poor tree health and numerous tree death, particularly on the sandy soils in southern Horqin (42°43′ N, 122°22′ E, our study area) (Jiao, 2001; Zhu et al., 2008) (**Fig. 1**).

[Figure]

**Figure 1** Location and environment of Horqin region in Liaoning province, China (study area)

A key driver for the degradation in water-limited ecosystems is regional low and erratic precipitation, which reduces soil water availability (Mereu et al., 2009). In semi-arid southern Horqin, three main soil-water related factors causes the degradation, i.e. the high inter-annual variation in precipitation, the high intra-seasonal variability in precipitation and the declining groundwater table (Jiao, 1989; Song et al., 2014). The forest's sensitivity to drought is highly species-specific, climate-specific and site-specific. It was reported that more than 85 % of roots system of Mongolian Scots pine grows in the upper 0.4 m soil layer in this region, and the root density is sharply decreased below 1.0 m soil depth (Su et al., 2006). However, it remains unclear how, and to what extent, these three factors are responsible for the degradation of this shallow-rooted species recorded in this region.

In this study, we hypothesize that on the long-period drought in semi-arid sandy environment raising from the high vari- ation of precipitation and low water holding capacity is the major reason causing the degradation of forest. The aims of this study were: (1) to compare the change of daily sap flux of MP based on sap flux density measurements in relation to the three contrasting precipitation years; (2) to determine the relationship between soil water availability, groundwater table and the responsibility of transpiration on driving from atmosphere; (3) to explore the effect of the severity of the drought stress over number of cycles of soil wetting and drying on daily sap flux and recovery ability.

[Figure]

**Figure 2** Annual variations of precipitation (a), mean air temperature (b) at Zhanggutai. Grey color indicates the data before the experiment (1983-2012) and black for the years of experiments (2013-2015). The dashed lines indicate the linear regressions over the whole period.

**2   Materials and methods**

**2.1  Site description**

The trial was carried out at the Zhanggutai National Desertification Control Trial Station located at the southern edge of the

Horqin region in Liaoning province, China (122° 22′ E, 42° 43′ N, at 226.5 m a.s.l.) (**Fig. 1**). The experiment was conduced in a 40 ha plantation with 35-year old MP. Tree density was 625 trees per ha. Management interventions and other human dis- turbances were limited by the installation of a secure fence around the experiment field. The site has a semi-arid, continental climate with a mean annual temperature of 7.9 °C, a frost-free period of 150–160 days per year, a mean annual evaporation of

1553 mm and a long-term annual mean precipitation of 475 mm ($P_{ave}$) over the last 30 years (1983–2012) with coefficient of variance of 0.27 (Zhu et al., 2005). Over the long period, there have been a number of consecutive dry years. For instance, annual precipitation between 1996 and 2004 were below $P_{ave}$ (**Fig. 2a**). Usually, about 60 to 70 % of annual rainfall occurs in the three months from June to August. The value of annual temperature over the last 30 years was increased slightly at a rate of 0.03 °C yr$^{-1}$ (**Fig. 2b**), while annual precipitation was slightly decreased with a rate of 2.0 mm yr$^{-1}$ (**Fig. 2a**). The soil is sandy with a sedimentary aeolian sand layer more than 3 m and an ancient alluvial sand layer with the total depth more than

126 m (Jiao, 1989). The mean bulk density of the upper 2 m soil layer is 1.61 g cm$^{-3}$. The mean soil texture is 83 % of sand (>

0.05 mm), 9 % of silt (0.05–0.002 mm) and 8 % of clay (< 0.002 mm). The organic matter content is 0.3–1.0 g kg$^{-1}$. The understory plant species in the forestry are *Acer pictum* subsp. *mono* Maxim, *Crataegus pinnatifida var. major N. E. Brown.*,

*Lespedeza bicolor* Turcz., *Artemisia halodendron* Turcz et Bess., *Cleistogenes chinensis* Maxim.

**2.2 Experimental design and samplings**

To break the prevailing northerly winds, the MP were planted in a square-grid pattern with 4 m for both row spacing and plant distance. Total area of experiment was 400 m$^2$ (20×20 m) containing 25 trees. All trees in the area were planted at same year in sole system surrounded by a wire fence (**Fig. 3**). The growth of trees in the experiment was normal in 2013, however, the leaves of trees in 2015 turned to grey slightly. The obvious defoliation or death did not occurred in 2015. Sap flow sensors were installed in each tree (totally 25 trees) in experimental area in 2013. Due to the damage of sensors, 22 left in 2014 and 13

left in 2015. The characteristics of the sampled trees are shown in **Table 1**. Diameters at breast height (DBH) were measured with a diameter tape and tree height with an altimeter. The thickness of bark, sapwood and heartwood were measured by sampling core at the height of sensors installed with a Pressler increment borer. Thickness measurements were made with a

Vernier caliper with tissue boundaries identified based on color. In our Mongolian Scots pine, the sapwood color was yellow- white and that of the heartwood was tan. A few drops of methyl orange solution helped define the interface where the boundary was indistinct. The DBH, tree height and sapwood areas of the sampled trees in 2013, 2014 and 2015 were not significantly different ($P > 0.05$), indicating a good uniformity of testing trees.

[Figure]

**Figure 3** Sketch map of 25 sample trees (stars) planted in a 4×4 m spaced square grid of about 400 m$^2$ (dashed line is border fence). Tree ages were identical and tree sizes were similar. The number of instrumented trees decreased in 2014, and again 2015 due to sensors damage during the reinstallation. Details of samples see **Table 1.**

**Table 1** Diameter at breast height (DBH, cm), tree height ($H$, m), height of first live branch ($H_b$, m), 1$^{st}$ quartile of DBH ($Q_1$),

3$^{rd}$ quartile of DBH ($Q_3$), sapwood width (SW, cm), sapwood area ($A_s$, cm$^2$) in 2013 to 2015. The mean values and standard deviations (S.D.) were given, the $n$ is the number of sampling trees.

| Year | DBH (cm) | | | $H$ (cm) | $H_b$ (cm) | SW (cm) | $A_s$ (cm$^2$) |
|---|---|---|---|---|---|---|---|
| | mean ± S.D. | $Q_1$ | $Q_3$ | | | | |
| 2013 ($n = 25$) | 18.0 ± 2.7 | 16.1 | 18.9 | 10.3 ± 0.7 | 4.5 ± 1.1 | 5.5 ± 0.6 | 203 ± 58.6 |
| 2014 ($n = 22$) | 17.1 ± 2.1 | 15.6 | 18.7 | 9.3 ± 0.8 | 3.7 ± 0.5 | 5.3 ± 0.5 | 182 ± 42.0 |
| 2015 ($n = 13$) | 17.7 ± 2.2 | 17.2 | 18.8 | 9.1 ± 0.8 | 3.7 ± 0.3 | 5.4 ± 0.5 | 189 ± 52.3 |

**2.3 Measurements**

**2.3.1 Micrometeorological variables**

Micrometeorological variables including solar radiation ($R_s$), net radiation ($R_n$), air temperature ($T_a$), relative humidity (RH), wind speed ($W_s$) and rainfall ($R$) were measured using an automatic weather station (AR5, Avalon Scientific, Inc. USA) located about 50 m away from the experimental field. All sensors were installed 2.0 m above the ground except the rain gauge, which was 0.5 m above the ground. Variables were measured at 1 min intervals, averaged and recorded per hour. Reference evapotranspiration ($ET_0$) was calculated using the FAO Penman-Monteith equation based on the variables $R_n$, $T_a$, RH and $W_s$

(Allen et al., 1998) at hourly base (Eq. (1)). Daily $ET_0$ was summed from hourly $ET_0$ for a day.

$$ET_0 = \frac{0.408(R_n - G) + \gamma \frac{900}{T_a + 273} u_2 (e_s - e_a)}{\Delta + \gamma(1 + 0.34 u_2)} \qquad (1)$$

where $ET_0$ = reference evapotranspiration (mm h$^{-1}$),

$\Delta$ = slope of saturated water vapour pressure against air temperature $T_a$ (kPa °C$^{-1}$),

$R_n$ = net radiation (MJ m$^{-2}$),

$G$ = soil heat flux (MJ m$^{-2}$),

$\gamma$ = the psychrometric constant (kPa °C$^{-1}$),

$e_s$ = saturated vapour pressure (kPa),

$e_a$ = actual vapour pressure (kPa), and

$u_2$ = mean wind speed at 2 m height (m s$^{-1}$).

The value of vapor pressure deficit ($D$, kPa) was calculated using the following formula (Campbell and Norman, 1998):

$$D = 0.611 exp^{\left(\frac{17.502 T_a}{T_a + 240.97}\right)}(1 - \text{RH}) \qquad (2)$$

**2.3.2 Soil moisture content and groundwater table**

Volumetric soil moisture contents ($\theta$, %) were measured at depths of 0.1, 0.2, 0.4, 0.6, 0.9, 1.2, 1.6 and 2.0 m using ECH$_2$O

EC-5 sensors (Decagon Devices Inc., USA). Three placements in experiment area were measured. Each placement was set between four neighborhood sample trees. Measurements were done at 10 min intervals with hourly means recorded by a

SQ2020 data logger (Grant Instruments Ltd, UK). The sensors was calibrated using a site-specific equation based on the oven-drying method (Eq. (3)):

$\theta = 0.9677\theta_s + 0.2635$  $(R^2 = 0.96, n = 194, \text{RMSE} = 0.41)$  (3)

where $\theta_s$ are the output of the sensors; $\theta$ is the calibrated soil moisture content at each depth and placement. The mean $\theta$

within a certain soil layer was weight-averaged based on the depth of sensor installation. The mean field capacity ($\theta_{fc}$) in 0-1

m soil layer of testing soil is 18.0 % by field observation. The minimum soil moisture content ($\theta_{min}$) measured during three years was 2.3 %. Relative extractable water (REW) in the upper 1.0 m soil layer was calculated using Eq. (4) (Granier,

1987).

$\text{REW} = \frac{\overline{\theta}_{0-1.0\ m} - \theta_{min}}{\theta_{fc} - \theta_{min}}$  (4)

The more specific classification to quantify the degree of drought at our site is defined in **Table 2**.

Groundwater table ($g_w$) was monitored *in situ* manually once per month using a measuring tape with a cone.

**Table 2** Classification of soil drought based on relative extractable water (REW) from the measurements and bio-physiological traits from preliminary reports in Mongolian Scots pine. The $T_r$ indicates transpiration rate, $C_s$ for stomatal conductance and $C_i$

for intercellular carbon oxide concentration.

| Parameter | Volumetric soil moisture content (%) | REW | Degree of drought | Description of bio-physiological traits |
|---|---|---|---|---|
| $D_0$ | $\overline{\theta}_{0-1.0\ m} > 0.4\ \theta_{fc}$ | REW > 0.31 | No drought | Normal growth |
| $D_{mil}$ | $0.3\ \theta_{fc} < \overline{\theta}_{0-1.0\ m} \leq 0.4\ \theta_{fc}$ | $0.20 < \text{REW} \leq 0.31$ | Mild drought | Weak growth (Jiao, 2001); $T_r$, $C_s$ and $C_i$ decreased by 46.2 %, 33.2 % and 0.9 %, respectively (Zhu et al., 2005; Tang et al., 2015); |
| $D_{mod}$ | $0.2\ \theta_{fc} < \overline{\theta}_{0-1.0\ m} \leq$ | $0.08 < \text{REW} \leq 0.20$ | Moderate drought | 30% leaves withered (Zhu et al., 2005); $T_r$, $C_s$ and |

| | 0.3 $\theta_{fc}$ | | | $C_i$ decreased by 62.1 %, 48.6 % and 51.1 %, respectively (Zhu et al., 2005; Tang et al., 2015); |
| --- | --- | --- | --- | --- |
| $D_s$ | $\overline{\theta}_{0-1.0\ m} \leq 0.2\ \theta_{fc}$ | REW ≤ 0.08 | Severely drought | Leaves withered and some of the branch die (Jiao, 2001); $T_r$, $C_s$ and $C_i$ decreased by 70.9 %, 77.3 % and 67.6 %, respectively (Zhu et al., 2005; Tang et al., 2015) |

**2.3.3 Sap flux density measurements**

Sap flux density in the outermost sapwood (0–3 cm depth) ($J_{s\text{-outter}}$, cm min$^{-1}$) was measured continuously using the Granier-type thermal dissipation method (Dynamax Inc., Houston. TX. USA). Each probe was installed under the cambium on the north side of the stem at breast height (1.3 m) with pairs of probes 0.04 m apart vertically. The upper probe included a heater and the lower probe was unheated and so remained at trunk temperature for reference. Each sensor was carefully removed at the end of each growing season (in November) and reinstalled at the initial stage in next year (in April). The temperature difference between the upper (heated) probe and the lower (reference) probe was measured at 1-min intervals, with mean values recorded at 10-min intervals using SQ2020 data loggers. The sensors were shielded with thick aluminum-faced foam to minimize warming by radiation and exposure to rain and physical damage. The Granier empirical equation for $J_s$ was adopted as Eq. (5):

$$J_{s-\text{outter}} = 119 \times 10^{-4} \left( \frac{\Delta T_0 - \Delta T}{\Delta T} \right)^{1.231} \tag{5}$$

where $\Delta T$ is the actual temperature difference observed between heated and reference probes, and $\Delta T_0$ is the maximum $\Delta T$ value when sap flow is close to zero (generally just predawn) determined over about 10 consecutive days by twice linear regression (Lu et al., 2004; Dang et al., 2014).

For the sap flux density at inner part of sapwood (beyond 3 cm depth) ($J_{s-\text{inner}}$) which is much low due to the relative inactivity of xylem, we adopted a coefficient 0.56 from Scots pine (*P. sylvestris*) (Lu et al., 2004; Nadezhdina et al., 2002) to estimate the sap flux density at inner part of sapwood.

**2.3.4 Calculation of sap flux**

Volumetric sap flux ($J_t$, $cm^3 h^{-1}$) were the product of sap flux density and corresponding sapwood area on hourly scale. At first, the sap flux density measurements $J_s$ was converted to a daily base (Eq. (6)). The daily mean $J_t$ for all measured trees in experiment area was then used to calculate daily transpiration of the stand (Eq. (7)). Because all trees were at the same age and regularly-spaced, each tree was assumed to occupy equal ground per sapwood area. Hence, the ground area fraction of each tree ($A_{g,i}$, $cm^2$) was approximated as the product of individual sapwood area and the ratio of total stand sapwood area ($A_{s\text{-}stand}$, $m^2$) divided by total stand ground area ($A_{g\text{-}stand}$, $m^2$).

$$J_{t,ij} = (J_{s-outter,ij} \times A_{s-outter,i} + J_{s-inner,ij} \times A_{s-inner,i}) \times 60 \qquad (6)$$

and upscaling to stand transpiration ($T_s$, mm $day^{-1}$)

$$T_s = \frac{A_{s-stand}}{A_{g-stand}} \frac{1}{\sum_{i=1}^{n} A_{s,i}} \sum_{i=1}^{n} \sum_{j=1}^{24} J_{t,ij} \qquad (7)$$

where,

$J_{t,ij}$ is hourly sap flux of a tree $i$ at $j$ hour of a 24-hour period, $J_{s\text{-}outter,ij}$ is the mean sap flux density in the probe-touched sapwood of a tree $i$ at $j$ hour in a day, $J_{s\text{-}inner,ij}$ is probe-untouched part, $J_{t,ij}$ is the sap flux of a tree $i$ at $j$ hour in a day, $A_{s-outter,i}$

and $A_{s-inner,i}$ is sapwood area of the outter 3 cm width and the rest sapwood of tree $i$, respectively, $A_{s,i}$ equals $A_{s\text{-}outter,i}$ and $A_{s\text{-}inner,I}$

$n$ is the numbers of trees measured each year. $A_{s\text{-}stand}$ is total sapwood area of 25 sample trees ($m^2$), $A_{g\text{-}stand}$ is total ground area of the plot ($A_{g\text{-}stand}$, $m^2$).

**2.4 Statistical analyses**

The normalized $T_s$ dividing by the maximum over the whole experiment period was used for comparison and further analy- sis. The effect of soil moisture content on normalized $T_s$ and the ratio $T_s / ET_0$ was tested by one-way analysis of variance (ANOVA) and a Tukey HSD *post hoc* multiple comparisons test using SPSS 20 (SPSS Inc., Chicago, IL, USA). Significant correlations between normalized $T_s$ or the ratio $T_s / ET_0$ and environmental factors over different periods were determined by

Pearson's correction coefficient tests at $P < 0.05$ or 0.01. The other statistical analyses and plots employed OriginPro 2016

version 9.3 (OriginLan Inc., Northampton, MA, USA).

**3   Results**

**3.1   Seasonal dynamics of normalized $T_s$ and environmental factors**

The amounts of rainfall during the investigation periods were 554 mm in 2013, 384 mm in 2014 and 408 mm in 2015, indicating a great variation among years. Meanwhile, rainfall in a year was concentrated over quite short periods (**Fig. 4a**) which also induced great inter-month variation of water supply in this region.

Daily soil moisture content exhibited large variances. The heterogeneity of soil moisture content with soil depth was significant ($P < 0.01$). In a wet year 2013, soil moisture content was higher than a dry year 2014 (**Fig. 4b**). In the second dry year 2015, there was a long drought period in July and August. After a heavy rainfall in late August (DOY 231 ), the soil of both upper and deep layers were refilled with a high soil moisture content. Based on our classifications (**Table 2**), the days of moderate and severe drought ($D_{mod} + D_s$) accounted for 19 %, 34 %, 66 % and 85 % of the whole three-year period for the four soil layers at 0–0.6, 0.6–1, 1–1.5 and 1.5–2 m, respectively. Thus, for MP it seems be the upper 1.0 m soil layer that provides the main water source, having the highest levels of soil moisture $\overline{\theta}_{0-0.6\,m}$ and $\overline{\theta}_{0.6-1\,m}$. Intense rainfall that infiltrated to and thus helped recharge the deeper layers of soil ($\overline{\theta}_{1.5-2\,m}$), were very rare from the later July in 2013 to later August in 2015.

There were no significant differences between years for daily $R_s$ ($P = 0.4$) or daily mean $D$ ($P = 0.25$) (**Figs. 4c, 4d**). The

$D$ over the whole period never exceeded 2.4 kPa (**Fig. 4d**). The groundwater table ($g_w$) at the start of the experiment was 5.2

m but significantly lowered to 5.6 m at the end of the experiment in 2015 (**Fig. 4e**).

The daily normalized $T_s$ showed similar seasonal patterns with $R_s$ (**Fig. 4f**), indicating the radiation was a major factor to affect plant transpiration. Overall, normalized $T_s$ was at a relative high level in May each year until August, and then gradually decreased to a low level in late October. The seasonal dynamics of normalized $T_s$ reflected the variations in physiological traits of MP and meteorological factors. The normalized $T_s$ between the years was significantly different ($P < 0.01$). The average daily normalized $T_s$ is 0.52 in 2013, decreased by 25 % in 2014 and further by 58 % in 2015. The maximum daily normalized

$T_s$ over three years occurred in 2013, the value of the maximum daily normalized $T_s$ in 2014 and 2015 is 0.68 and 0.58, respectively. (**Fig. 4f**). The decreasing normalized $T_s$ between seasons was partially due to less rainfall and soil water availa- bility, and probably also due to the plant recovery capability in relation to the permanent changes in plant physiological traits.

[Figure]

**Figure 4** Seasonal time courses of precipitation, mean volumetric soil moisture content ($\overline{\overline{\theta}}$), solar radiation ($R_s$), vapour pressure deficit ($D$), groundwater table ($g_w$), and normalized daily transpiration of stands ($T_s$) dividing by the maximum over the whole experiment period. The grey area in (b) indicates soil moisture under moderate and severe drought condition.

**3.2 Response of transpiration to atmosphere driving and changes with soil drought**

The ratio $T_s$ / $ET_0$ was used to reflect the response of transpiration to atmosphere driving. We found that $T_s$ / $ET_0$ under sufficient water supply condition ($D_0$) was about 0.29 ± 0.09 (**Fig. 5a**), 15% higher than under mild drought ($D_{mil}$), 62% higher than under moderate drought ($D_{mod}$) and 149 % higher than under severe drought ($D_s$). The maximum ratio of daily $T_s$ / $ET_0$ increased with

REW sharply at low REW but keep constant when the REW was above 0.31 (**Fig. 5b**). This indicated that there are the other factors besides the atmospheric and soil moisture affected the sap flux of MP, in which the stomatal regulation or the seasonal variation of biological rhythms is important one.

[Figure]

**Figure 5** The ratio $T_s$ / $ET_0$ affected by soil droughts. Normalized sap flux by using reference evapotranspiration indicates a potential tran- spiration ability under maximum evaporative demand caused by meteorological factors, the relationship between $T_s$ /$ET_0$ and relative ex- tractable water (REW) is mainly affected by plant traits. The maximum of $T_s$ / $ET_0$ at the REW step of 0.02(dimensionless) are selected out (red circles in (b)) and modelled by an exponential function. The dashed line is at REW=0.31. Values of $T_s$ / $ET_0$ followed by different letters are significantly different at $P < 0.05$ by univariate ANOVA (post hoc Tukey HSD).

**3.3 Progressive decline of normalized $T_s$ with developing of drought and recovery following rain**

The 49-day periods from DOY 203 to 251 each year was chosen to illustrate the changes in normalized $T_s$ with REW, a dry- wet shift. In the period of wet year (2013), the soil moisture is always in $D_0$ level (without water stress) and mean normalized

$T_s$ was about 0.57 (**Fig. 6a**). In the first dry year (2014), normalized $T_s$ decreased by as much as 18 % under a mild water stress ($D_{mil}$) and further by 40 % under moderate stress ($D_{mod}$). The normalized $T_s$ was greatly recovered after a heavy rain (**Fig. 6b**).

In the second dry year (2015), the normalized $T_s$ decreased by 73 % under $D_{mod}$ and further by 74 % under $D_s$ stage (**Fig. 6c**).

The daily normalized $T_s$ under $D_s$ was only 0.15. This very little transpiration level likely only sufficient to maintain the survival of MP. After a heavy rainfall, even the soil water status was improved a lot, the normalized $T_s$ of trees was still very low (less than 0.38), indicating the $T_s$ of MP was difficult to be recovered.

[Figure]

**Figure 6** The comparison of normalized daily transpiration ($T_s$) and relative extractable water (REW) in the upper soil layer (above 1 m)

during maximum growth period from DOY 203 to 251 in 2013 to 2015. $D_0$, $D_{mil}$, $D_{mod}$, and $D_s$ are no, mild, moderate and severe droughts stage, respectively (see Table 2). The obvious increase of REW was due to the heavy rainfall events which were seen in Fig.4.

**4 Discussion**

**4.1 Reduction of soil moisture content under droughts**

In our site, the long term trends for increasing air temperature and decreasing annual precipitation (**Fig. 2**) is unfavorable to the growth of trees. The declining groundwater and the coarse sandy soil (>83 % sand particles in our site) prevented capillary ascension efficiently (less than 0.5 m) (Vincke and Thiry, 2008). Sandy soils have low water holding capacity and high hy- draulic conductivity, thus water percolates through this soil quickly after a rain. During the three-year periods in our site, there are an effective rain event every 14 days averagely (rainfall intensity is more than 10 mm per times). Under well-wetted soil conditions ($D_0$), $\overline{\theta}_{0-1.0\,m}$ was depleted at the high rate of 1.9 vol % per day during the first two days and at the rate of 0.35 vol %

per day during the subsequent nine days (**Fig. 7**) because of either soil water holding capacity or great water uptake by trees.

The depleting rate of $\overline{\theta}_{0-1.0\,m}$ under the drought conditions was only 0.09 vol % per day under severe drought. That indicates the only little of water was absorbed by trees under severe water stress. Our results suggested the plant might adjust their physiological traits, e.g. closing stomatal and reducing root system to at first priority for the survival. The sap flux declines very quickly in desiccated root system (Mereu et al., 2009).

[Figure]

**Figure 7** There will be an effective rain event every 14 days averagely in our site (Rainfall intensity is more than 10 mm per times), which acted as a window to analyze the decrease rate of soil water during this period. Scatter-line plot described the relationships between decrease rate of upper soil moisture ($\overline{\theta}_{0-1.0\,m}$) and time under different initial degree of drought levels which were defined in **Table 2**.

**4.2 Contribution of water in the upper and deep soil layers**

The MP is a shallow-rooted species with root density decreasing sharply below 1.0 m (Jiang et al., 2002; Zhu et al., 2005; Zhu et al., 2008), implying the soil moisture in the upper 1.0 m layer provides major water source for transpiration (Su et al., 2006; Wei et al., 2013; Song et al., 2014). In our study, the rapid recovery of normalized $T_s$ when $\overline{\theta}_{0-1.0\,m}$ was increased after a rain (**Figs. 4 and 6**) suggested that MP was very sensitive to the changes in the available water in the upper soil layer. Uptake by the shallow roots decreased very significant as this soil layer dried out (**Fig. 6**). However, under severe drought, for example in August of 2015, the MP trees used quite amount of deep soil water. It might be carried out by developing more letaral root system in deep soil. The fine roots of Scots pine die quickly under drought conditions (Vanguelova and Kennedy, 2007). Therefore, it would cause a death of new developed fine root system, resulting a permanent declining in the capability of transpiring water even when the soil water status was improved (**Fig. 4**). The death of fine root in deep soil layer may explain why after a rainfall, post-stress sap flux recovery is very small after a long and severe drought in 2015.

**4.3 Groundwater is an important source for plant adaptation under long and severe drought**

Mongolian Scots pine is a dimorphic-rooted species where, the maximum taproot depth in a sandy soil can up to 2.7 m (Canadell et al., 1996), and even to 5.2 m for a 42-year-old tree in a sandy soil near our site (Jiang et al., 2002). Our results on the depletion of soil water in 1.5-2 m soil layers, existed but not large, also suggested a deep taproot depth in MP. The deep taproot enables trees to use deeper water source (i.e. groundwater), especially under drought (Barbeta et al., 2015; Hentschel et al., 2016). This is likely to occur when soil moisture content in the upper soil layers (0–60 cm) declines to 3.6 % (Wei et al., 2013; Song et al., 2016a). In our site, $g_w$ lowered from $5.03 \pm 0.14$ m in 2013 to $5.47 \pm 0.09$ m in 2015 (**Fig. 4e**). From late 2014, the value of $g_w$ was always far deeper than 5.2 m and thus unlikely accessible directly by our instrumented trees if their tap roots were shallower than 5.2 m. However, in the severe drought ($D_s$, with minimum $\overline{\theta}_{0-1.0\,m}$ as low as 2.3 %), we recorded a clear diurnal pattern of sap flux with the much reduced daily normalized $T_s$ ( 0.15, or 28.2 % of that for $D_0$). Hence, we inferred that significant groundwater contributions to $T_s$ occurred only under severe drought conditions though determining just what proportion of that water came from the groundwater or from tree storage is beyond the scope of this study. It has been reported that as rainfall decreases, tree dependence on groundwater increases (Kume et al., 2007).

[Figure]

**Figure 8** The ratio $T_s$ / ET$_0$ in Mongolian Scots pine decreased with the declining of the groundwater table ($g_w$)

**4.4 Transpiration of the plantation and implications**

There is a complex interplay between the various meteorological factors, e.g. solar radiation, vapour pressure deficit, air tem- perature, wind speed and relative humidity, and directly or indirectly influences transpiration in a tree. These variables were aggregated into a variable ET$_0$, which serves as an index of atmospheric water demand power (Allen et al., 1998; Zha et al.,

2010). Therefore, as expected, changes in ET$_0$ trigger a prompt plant response in terms of transpiration. Changes in precipita- tion (and hence soil moisture) affect transpiration but likely over a long temporal scale (Yan et al., 2016). Our results also showed a strong reduction in the ratio $T_s$ / ET$_0$ mostly after a long period drought. Using normalized transpiration allows to focus on the effects of soil water availability and plant physiological responses. This behavior has also been found in Scots pine in Europe (Poyatos et al., 2005), presenting the strong effects of stomatal regulation for controlling the rate of water loss.

The significant fall in $g_w$ seems to explain the difficulty in plant recovery of $T_s$ after a heavy rain.

Transpiration in a coniferous forest is often conservative with relatively low values of canopy conductance (Levitt, 1980). For instance, Scots pine has a rather conservative water use strategy with a very plastic response to intermittent dry periods with high use of stored water (Arneth et al., 2006; Verbeeck et al., 2007). In our study, we found MP was more moderate in its water consumption than many broad-leaved forest tree species growing nearby (e.g. *Populus* spp) (Zhu et al., 2005). Although the groundwater table decreases in our experiment, the MP still contributes less to the groundwater table decline than the more extensive and/or intensive agricultural land uses (0.1 m per year) (Song et al., 2016b). The lateral roots of an MP tree can extend laterally to about 0.65-times tree height (Jiang et al., 2002; Su et al., 2006). This helps MP to obtain water from the upper soil layers efficiently (Song et al., 2014). However, the ability of MP to maintain the normal water use level decreased greatly during prolonged severe drought, which implies the limitation of trees to climate change (Waromg et al., 1979), espe- cially when the extreme weather events increase in the future. Meanwhile, it was reported that the mature period of Mongolian

Scots pine occurred in advance when introduced from the north (origin distribution region) to south (planted region, this study area) (Jiao et al. 1989; 2001). Therefore, the difficulty in recovery for water uptake by 30 years MP trees under severe drought might also caused by the low growth vigor of old trees. It implies that the re-forestry might be necessary when MP trees are over 30 years old in the sandy site.

**5 Conclusions**

The mean stand transpiration of Mongolian Scots pine was high in wet year 2013, but decreased by 25% in dry year 2014 and further by 58% in second dry year 2015. The high inter-annual as well as the high intra-seasonal variability in precipitation induced the great fluctuation of soil moisture at the upper soil layer frequently, which brought the great effect on sap flux of this shallow-rooted species. The daily stand transpiration reduced with drought by as high as 74% as the duration and intensity of drought was high in dry years. The ability of recovery in plant transpiration following heavy rain, however, was limited greatly with the duration and severity of drought. Our results suggest that the degradation in MP plantation is attributable to the combined effects of large temporal variation in rainfall and the ability of specific recovery after the occurrence of drought.

The results could help farmer improve the management and sustainability of MP forestry by optimizing plant density and reforestation in semi-arid region.

**Acknowledgements**

[revised manuscript text omitted]

Nadezhdina, N., Cermak, J. and Ceulemans, R.: Radial patterns of sap flow in woody stems of dominant and understory species:

scaling errors associated with positioning of sensors, Tree Physiol., 22, 907-918, 2002.

Poyatos, R., Llorens, P. and Gallart, F.: Transpiration of montane *Pinus sylvestris* L. and *Quercus pubescens* Willd. forest stands measured with sap flow sensors in NE Spain., Hydrol. Earth Syst. Sc., 9, 493–505, 2005.

Song, L.N., Zhu, J.J., Li, M.C. and Yu, Z.Y.: Water utilization of *Pinus sylvestris* var. *mongolica* in a sparse wood grassland in the semiarid sandy region of Northeast China, Trees, 28, 971–982, 2014.

Song, L.N., Zhu, J.J., Li, M.C., Zhang, J.X. and Lv, L.Y.: Sources of water used by *Pinus sylvestris* var. *mongolica* trees based on stable isotope measurements in a semiarid sandy region of Northeast China, Agr. Water Manage., 164, 281–290, 2016a.

Song, L.N., Zhu, J.J., Li, M.C. and Zhang, J.X.: Water use patterns of *Pinus sylvestris* var. *mongolica* trees of different ages in a semiarid sandy lands of Northeast China, Environ. Exp. Bot., 129, 94–107, 2016b.

Su, F.L., Liu, M.G., Guo, C.J. and Zhang, Q.: Characteristics of vertical distribution of root system of Mongolian scots pine growing in sandy area and influence to the soil, Soil and Water Conservation in China, 286, 20–23, 2006 (in Chinese with

English abstract).

Su, H., Li, Y.G., Liu, W., Xu, H. and Sun, O.J.: Changes in water use with growth in *Ulmus pumila* in semiarid sandy land of northern China, Trees, 28, 41–52, 2014.

Tang, F.D., Lin, Y. and Li, Y.: Impact of water stress photosynthesis characteristics of Mongolian pine seedlings and Grafting

Korean pine seedlings with stocks of Mongolian pine, Journal of Liaoning University, 42, 274-276, 2015 (in Chinese with English abstract).

Vanguelova, E.I. and Kennedy, F.: Morphology, biomass and nutrient status of fine roots of Scots pine ( *Pinus sylvestris* ) as influenced by seasonal fluctuations in soil moisture and soil solution chemistry, Plant Soil, 270, 233–247, 2007.

Verbeeck, H., Steppe, K., Nadezhdina, N., De Beeck, M.O., Deckmyn, G., Meiresonne, L., Lemeur, R., Cermak, J., Ceulemans,

R. and Janssens, I.A.: Model analysis of the effects of atmospheric drivers on storage water use in Scots pine, Biogeosci- ences, 4, 657–671, 2007.

Vincke, C. and Thiry, Y.: Water table is a relevant source for water uptake by a Scots pine (*Pinus sylvestris* L.) stand: Evidences from continuous evapotranspiration and water table monitoring, Agr. Forest Meteorol., 148, 1419–1432, 2008.

Waromg, R.H., Whitehead, D. and Jarvis, P.G.: The contribution of stored water to transpiration in Scots pine, Plant, Cell &

Environment, 2, 309–317, 1979.

Wei, Y.F., Fang, J., Liu, S., Zhao, X.Y. and Li, S.G.: Stable isotopic observation of water use sources of *Pinus sylvestris* var.

*mongolica* in Horqin Sandy Land, China, Trees, 27, 1249–1260, 2013.

Wesche, K., Walther, D., Wehrden, H.V. and Hensen, I.: Trees in the desert: Reproduction and genetic structure of fragmented

*Ulmus pumila* forests in Mongolian drylands, Flora, 206, 91–99, 2011.

Yan, M.J., Zhang, J.G., He, Q.Y., Shi, W.Y., Otsuki, K., Yamanaka, N. and Du, S.: Sapflow-based stand transpiration in a semiarid natural oak forest on China's Loess Plateau, Forests, 7, 227–240, 2016.

Yuan, M.W., Zhang, L.Z., Gou, F., Su, Z., Spiertz, J.H.J., van der Werf, W.: Assessment of crop water productivity in semi- arid Inner Mongolia. Agr. Water Manage., 122, 28–38, 2013.

Zha, T.S., Barr, A.G., Kamp, G.V.D., Black, T.A., Mccaughey, J.H. and Flanagan, L.B.: Interannual variation of evapotran- spiration from forest and grassland ecosystems in western Canada in relation to drought, Agr. Forest Meteorol., 150,

1476–1484, 2010.

Zheng, X., Zhu, J.J., Yan, Q.L. and Song, L.N.: Effects of land use changes on the groundwater table and the decline of *Pinus*

*sylvestris* var. *mongolica* plantations in southern Horqin Sandy Land, Northeast China, Agr. Water Manage., 109, 94–

106, 2012.

Zhu, J. J., Li, F.Q., Xu, M.L., Kang, H.Z. and Wu, X.Y.: The role of ectomycorrhizal fungi in alleviating pine decline in semiarid sandy soil of northern China: an experimental approach, Ann. Forest Sci., 65, 304, 2008.

Zhu, J.J., Kang, H.Z., Li, Z.H., Wang, G.C. and Zhang, R.S.: Impact of water stress on survial and photosynthesis of Mongolian pine seedlings on sandy land, Acta Ecologica Sinica, 25, 2527-2533, 2005 (in Chinese with English abstract).

Zhu, J.J., Zeng, D.H., Kang, H.Z., Wu, X.Y. and Fan, Z.P.: Decline of *Pinus sylvestris* var. *mongolica* plantations on Sandy

Land, Chinese Forestry Press, Beijing, 264 pp, 2005 (in Chinese with English abstract).